# Targeting HIF-1α by Natural and Synthetic Compounds: A Promising Approach for Anti-Cancer Therapeutics Development

**DOI:** 10.3390/molecules27165192

**Published:** 2022-08-15

**Authors:** Rituparna Ghosh, Priya Samanta, Rupali Sarkar, Souradeep Biswas, Prosenjit Saha, Subhadip Hajra, Arijit Bhowmik

**Affiliations:** Department of Cancer Chemoprevention, Chittaranjan National Cancer Institute (CNCI), Kolkata 700026, India

**Keywords:** HIF-1α, hypoxia, metastasis, angiogenesis, cancer stem cells, natural compounds, synthetic drugs

## Abstract

Advancement in novel target detection using improved molecular cancer biology has opened up new avenues for promising anti-cancer drug development. In the past two decades, the mechanism of tumor hypoxia has become more understandable with the discovery of hypoxia-inducible factor-1α (HIF-1α). It is a major transcriptional regulator that coordinates the activity of various transcription factors and their downstream molecules involved in tumorigenesis. HIF-1α not only plays a crucial role in the adaptation of tumor cells to hypoxia but also regulates different biological processes, including cell proliferation, survival, cellular metabolism, angiogenesis, metastasis, cancer stem cell maintenance, and propagation. Therefore, HIF-1α overexpression is strongly associated with poor prognosis in patients with different solid cancers. Hence, pharmacological targeting of HIF-1α has been considered to be a novel cancer therapeutic strategy in recent years. In this review, we provide brief descriptions of natural and synthetic compounds as HIF-1α inhibitors that have the potential to accelerate anticancer drug discovery. This review also introduces the mode of action of these compounds for a better understanding of the chemical leads, which could be useful as cancer therapeutics in the future.

## 1. Introduction

Inadequate oxygen supply to tumor tissues creates hypoxia, one of the most significant clinical conditions responsible for cellular proliferation, angiogenesis, metastasis, and propagation of cancer stem cells (CSCs). In the case of hypoxia, oxygen concentration in the normal human renal cortex and brain tissues changes from 9.5% to ~1.3% and ~4.6% to ~1.7%, respectively [1,2]. However, an adequate amount of oxygen and nutrition supply is highly needed for the growth of any multicellular organ, as well as solid tumors. However, oxygen supply becomes restricted when distance between a tumor cell and blood vessel exceeds 70 μm [3]. This condition is known as diffusion-limited hypoxia. Additionally, chronic hypoxia leads to a restricted supply of oxygen, which depends on severe structural and functional abnormalities of the tumor microvessels (perfusion-limited O_2_ delivery), a deterioration of the diffusion geometry (diffusion-limited O_2_ delivery), tumor-associated, and therapy-induced anemia [4]. In hypoxic conditions, reactive oxygen species (ROS) production increases in tumor cells [5]. Generation of ROS triggers damage of cellular biomolecules such as lipids, proteins, DNA, and RNA that leads to cell death [6]. Despite this, cancer cells find smart ways to escape cell death and ultimately survive in the unfavorable conditions [7].

The most common survival features of cancer tissues include their augmentation toward metabolic reframing, angiogenesis, and metastasis. These responses are mediated by hypoxia-inducible factor (HIF) family of proteins, the expression of which becomes altered upon hypoxia. The HIF pathway remains conserved from *Caenorhabitis elegans* to human and becomes activated to maintain oxygen homeostasis in hypoxic conditions [8]. Hypoxia-inducible factor was first discovered by Semenza and coworkers, and they described the presence of hypoxia response element (HRE), which is conserved by every target gene of HIFs [9]. HIF is a heterodimeric protein with a basic helix loop helix structure that contains one α and one β subunit. In addition to the ubiquitously expressed HIF-1, the HIF family consists of two other members called HIF-2 (also called EPAS1, MOP2, or HLF) and HIF-3. HIF-1 and HIF-2 share many functional and structural similarities such as DNA recognition, binding and heterodimerization with HIF-1β (ARNT). HIF-2α is a constitutively expressed cytoplasmic protein and mainly found in vascular endothelial cells. Various studies reported that it is also up-regulated in cancer cells. Another member of this family, HIF-3α, functions mainly as a negative regulator of HIF-1α. An alternatively spliced variant of HIF-3α called IPAS directly binds to HIF-1α and prevents its binding with HIF-1β which results in the formation of an abortive complex. In this way, HIF-3α prevents HIF-1α from forming an active transcription factor and in turn hinders binding of HIF-1α to the HRE sequence of hypoxia-induced genes [10]. Multiple studies reported the prominent role of HIF-1α in malignant cells. Hence, in this review, we present insights into the complexity of cellular metabolism, metastasis, tumor angiogenesis, and survival of cancer stem cells induced by HIF-1α.

The role of HIF-1α in the apoptosis of cancer cells is controversial. In some types of cancer, HIF-1α triggers apoptosis, whereas in some cases it induces cell proliferation. Usually, normal as well as cancerous cells die owing to the depletion of oxygen. In certain cases, HIF-1α triggers apoptosis in cancer cells via regulation of pro-apoptotic, anti-apoptotic and apoptotic proteins. However, sometimes, HIF-1α also arrests cell growth in the G1/S phase to escape cell death [11]. Consequently, activation of cell cycle arrests as well as induction of apoptosis encourages cancer cells to be resistant in radiation and chemotherapy.

Along with this, in the case of angiogenesis, hypoxia triggers the formation of new blood vessels as well as sprouting from existing blood vessels to increase the supply of oxygen. Expression of angiogenic markers such as VEGF, TGF-β, PDGF-β, plasminogen activator-1 (PAI-1), erythropoietin (EPO) and GLUT-1 are regulated by HIF-1α [12]. HIF-1α is known to be involved in all steps of blood vessel formation. However, tumor vasculature has different morphological features from normal blood vessels [12]. Hypoxia induces the imbalance between pro and anti-angiogenic factors production, which leads to enhanced, rapid, and chaotic blood vessel formation [13,14]. However, other than involvement in angiogenic progression HIF-1α also plays an important role in other hypoxia-induced hallmarks of cancer, such as induction of metastasis [15,16]. HIF-1α regulates a broad range of genes involved in epithelial to mesenchymal transition, disrupting basement membrane of surrounding tumor tissue and invasion [16,17]. In this way, hypoxia helps the cells of the primary tumor to escape the hypoxic region and migrate to a distal site to form a secondary tumor by regulating HIF-1α. In addition, adaptations of these new characteristics make a cancer cell more aggressive in hypoxic conditions.

It is reported that hypoxia leads to clonal selection of tumor cells [18,19]. In hypoxic conditions, HIF-1α promotes enrichment of cancer stem cells (CSCs) to adopt a different way to survive [19]. CSCs are a subpopulation of cells within tumor. These cells have self-renewing as well as self-differentiation properties [20]. In addition, CSCs have altered gene expression to gain drug resistant properties [20]. Various cell surface markers such as CD44, CD24, CD29, CD90, CD133, epithelial-specific antigen (ESA), and aldehyde dehydrogenase1 (ALDH1) have been used to isolate and recognize CSCs from different tumors [21]. Although the mechanism of hypoxic controls of cancer is not fully revealed, it is reported that CSCs usually reside in the hypoxic region of the tumor [22]. Danet et al., cultured human hematopoietic stem cells (HSCs) under hypoxic conditions and showed that they can promote their ability to repopulate when they are transplanted to nonobese diabetic (NOD)/severe combined immunodeficiency (SCID) mice [23]. Ezashi et al., reported that embryonic stem cells are maintained by low oxygen tension and significantly block spontaneous cell differentiation [24]. However, recent studies suggested that hypoxia directly regulates cancer stem cell-related pathways [25,26]. Low oxygen tension may convert non-stem cancer cells into cancer stem cell-like status with increased self-renewing capacity as well as induction of essential stem cell factors, such as OCT4, Nanog, and c-MYC [25]. HIF-1α is critical for cancer stem cell maintenance, as knockdown of HIF-1α in cancer stem cells leads to reduce self-renewal capacity, increases apoptosis, and attenuates tumorigenesis [19,26].

Therefore, targeting HIF-1α can be an effective way to control various crucial hallmarks of cancer. Several drugs have already been designed to combat cancer by targeting HIF-1α. Some of these drugs are in clinical trial whereas, some show great potential in vitro and in vivo. Along with the synthetic chemotherapeutic drugs, some natural compounds have high probability to block HIF-1α pathways [27]. Henceforth, a brief description of different potential anti-cancer natural compounds and synthetic chemotherapeutic agents targeting HIF-1α are discussed in this review.

## 2. Structure and Regulation of HIF-1α

HIF, a heterodimeric protein, is comprised of two subunits, i.e., HIF-1α and HIF-1β, with a molecular weight of 120 kDa and 91–94 kDa, respectively [28]. HIf-1α contains a basic helix loop helix (BHLH) and PAS domain, which is named after three proteins first recognized in drosophila, i.e., Per, ARNT, and Sim. Both HIF-1α and HIF-1β possess PAS domain which is situated before BHLH domain near the N-terminal end. Moreover, HIF-1α contains three other important domains such as N-terminal transcriptional activation domain (NAD), C terminal transcriptional activation domain (CAD) and an oxygen dependent domain (ODD). HIF-1α binds to DNA through the basic domain of BHLH, whereas HLH domain is essential for dimerization of HIF-1α and HIF-1β [29]. In cells, HIF-1α is tightly regulated depending on the availability of oxygen, whereas HIF-1β (also known as ARNT) is constitutively expressed and remains abundant irrespective of alterations in oxygen tension. Furthermore, the stability of the HIF-1α protein mainly depends on the post translational alterations [30]. In normoxic conditions, HIF-1α possesses a pair of sequences in the C terminal portion of the protein resulting a shorter half-life and instability [31,32]. In the presence of oxygen, proline hydroxylase (PHD) promotes hydroxylation of two conserved proline residues (Pro402 and Pro564 in human) present in LXXLAP motif of ODD [30]. As a result of this hydroxylation, the von Hippel-Lindau protein (pVHL) recognizes and binds to HIF-1α, which in turn facilitates its degradation by poly ubiquitination [33]. In contrast, in hypoxic conditions, PHD becomes inactive and HIF-1α protein escapes the degradation. Hence, HIF-1α translocates into the nucleus followed by dimerization with ARNT and becomes activated to function as an effective transcription factor [34,35]. Along with these, p300 and CREB-binding protein (CBP) act as important transcriptional regulators of HIF-1α and in hypoxic conditions, it binds to the carboxy-terminal transactivation domain (CTAD) of HIF-1α. In the presence of oxygen, FIH1 (factor inhibiting HIF1) hydroxylates the asparagine residue at 803 positions in the CTAD region. This prevents the binding of p300/CBP in this region which in turn helps in the binding of pVHL–elongin–cullin-2 complex to ODD that augments proteasomal degradation of HIF-1α by 26S proteasome (Figure 1) [36].

Along with the regulation via ubiquitin-proteasome pathway, several growth factors are responsible for translational regulation of HIF-1α, such as insulin, IGF-1, IGF-2, EGF, v-SRC, endothelin-1, ADM, erythropoietin, and cytokines. These growth factors can induce synthesis of HIF-1α protein via PI3K/AKT/mTOR pathway by binding at the Tyrosine kinase receptor. PI3K activates its downstream regulator mTOR via AKT. mTOR phosphorylates eukaryotic translation initiation factor 4E (eIF-4E) binding protein (4E-BP1) and disrupt integrity of eIF-4E which is required for inhibiting cap-dependent mRNA translation of HIF-1α. In addition, mTOR promotes phosphorylation of the ribosomal protein S6, and induces HIF-1α translation (Figure 1) [9,37]. Bypassing proteasomal degradation facilitates HIF-1α translocation into the nucleus and formation of heterodimer with HIF-1β. This heterodimer specifically binds to a 5′-RCGTG-3′ hypoxia-responsive element (HRE) sequence in the promoter or enhancer of various hypoxia-inducible genes, which includes erythropoietin, vascular endothelial growth factor, glucose transporters, and glycolytic enzymes, as well as genes involved in iron metabolism, and cancer cell and cancer stem cell survival [38].

## 3. Role of HIF-1α in Cancer Progression

In response to both hypoxic stress and oncogenic signals, HIF-1α becomes activated and controls different mechanisms involved in cancer cell survival and proliferation resulting in the formation of vascular tumors with metastatic potential [39]. Other than these, a variety of mechanisms regulate HIF-1α mediated cancer stem cell propagation and maintenance [40]. For better understanding role of HIF-1α in cancer cell metabolism, angiogenesis, metastasis, and survival of cancer stem cells are discussed here.

### 3.1. Role of HIF-1α in Cellular Metabolism

Lowering the oxygen level in cancer cells not only causes HIF-1α gene over expression but also influences the fluctuation of cellular metabolic homeostasis [41,42]. Expression of HIF-1α affects the rate of different metabolic pathways such as glycolysis, glycogenolysis, neoglucogenesis, β-oxidation, citric acid cycle, etc. [43,44,45,46]. On the other hand, through these metabolic alterations, HIF-1α promotes insulin resistance and obesity in most of the cancer patients [47,48]. In hypoxia, HIF-1α acts as a transcription factor and regulates oncogenic metabolism by two ways, i.e., by promoting anaerobic glycolysis and by suppressing TCA cycle or mitochondrial oxygen consumption [49]. HIF-1α induces some glycolytic enzymes and transporter-like aldolase A, mitochondrial pyruvate kinase 1 (PDK1) and glucose transporters (GLUTs) [50,51,52,53]. In the cancer cells, HIF-1α not only stimulates the induction of GLUT1, GLUT3 and GLUT4 transporter to uptake blood glucose, but also enhances glycolytic breakdown of intracellular glucose by transactivating phosphofructokinase 1 (PFK1) and aldolase (Figure 2) [52,54,55,56].

Another glycolytic pathway member, Hexokinase 2(HK2), is a rate limiting enzyme involved in phosphorylation of glucose to glucose-6-phosphate and also reported to be up-regulated by HIF-1α in hypoxic cancer cells [57,58]. In addition to these, HIF-1α can trigger the expression of lactate dehydrogenase A (LDHA) which leads to conversion of glycolytic end products pyruvate and NADH to NAD+ and lactate [59]. This NAD+ is needed for another cycle of glycolysis which is essential for oncogenic metabolism. It is also evident that in cancer cells HIF-1α induces the expression of monocarboxylate transporter 4 MCT4, a plasma membrane binding transporter, which is known for its lactate extruding activity followed by pyruvate to lactate conversion [60,61]. The other mechanism of metabolic alteration by HIF-1α is the down-regulation of pyruvate dehydrogenase (PDH) expression by pyruvate dehydrogenase kinase 1 (PDK1) phosphorylation which successively blocks TCA cycle [62]. Generally, PDH mediates the breakdown of pyruvate to acetyl-CoA and CO_2_ in mitochondria. Hence, glycolysis dependency of cancer cells increases due to inhibition of PDH mediated breakdown of pyruvate to acetyl-CoA and CO_2_ in mitochondria [62]. Alteration of lipid biosynthesis is another approach of HIF-1α mediated metabolic regulation of cancer cells. In mitochondria, pyruvate produces acetyl-CoA that converts to citrate and translocates to cytoplasm and produces acetyl-CoA and oxaloacetate by the enzyme ATP citrate lyase. Although this acetyl-CoA is used for lipid biosynthesis in endoplasmic reticulum in normoxic conditions, its production is hindered by HIF-1α [62]. Therefore, for cancer cells, it is essential to expedite metabolic functions using alternative sources of fatty acid precursors. Hence, uptake of extracellular fatty acid is prompted by HIF-1α dependent activation of peroxisome proliferator activated receptor gamma (PPARγ) gene [46]. Activation of this gene promotes glycerolipid biosynthesis in cancer cells by inducing genes such as fatty acid binding proteins (FABP) 3, 4, and 7 [63]. Apart from these, phenomena reversion of TCA cycle occurs using HIF-1α induced isocitrate dehydrogenase (IDH) in cancer cells to maintain de novo synthesis of fatty acid [64]. Along with these, HIF-1α is also able to up-regulate glutaminase 1 (GLS1), an essential enzyme for glutamine metabolism that controls production of α-ketoglutarate, which is a main precursor molecule of de novo fatty acid synthesis [65]. Moreover, reduction in mitochondrial biogenesis to regulate oxygen consumption in cancer cell is a significant part of HIF-1α mediated metabolic alteration. This process of mitochondrial biogenesis requires transcription factor A (TFAM), which has a direct trans-activator such as MYC and MAX. However, MXI1, a member of MYC family and a negative regulator of MAX and MYC, is up-regulated by HIF-1α (Figure 2). This leads to suppression of mitochondrial biogenesis and lower oxygen consumption in HIF-1α over expressed cancer cell [66]. Hence, from this part of the review, it is evident that HIF-1α has significant role in metabolic regulation as well as survival and growth of cancer cells.

### 3.2. Role of HIF-1α in Regulating Angiogenesis

HIF-1α expression is the key mediator of angiogenesis in physiological conditions as well as patho-physiological conditions. As with the development of any multicellular organism, tumor development also depends on the adequate amount of oxygen and nutrients supply through the blood vessels. In tumor proliferation beyond 1–2 mm, oxygen and nutrients cannot diffuse properly to the core, which creates hypoxia. This hypoxic condition triggers activation of HIF-1α resulting in the overexpression of HIF regulated genes such as VEGF, metalloproteinases, chemokines to stimulate angiogenesis, and endothelial cells recruitment (Figure 2) [67,68,69]. HIF-1α expression is also regulated by mutations in tumor suppressor genes, such as VHL, p53, and PTEN [70,71,72] Moreover, overexpression of oncogenes that includes v-SRC, EGFR, HER2, and subsequent signaling through the phosphatidylinositol-3-kinase (PI3K) and mitogen-activated protein (MAP) kinase pathways activate HIF-1α expression [52,53,54]. In the above cases, HIF-1α ultimately promotes angiogenesis by regulating broad range of genes including VEGF [73,74]. Deficiency in HIF-1β/ARNT in hepatoma cells results in a less vascular, slow growing tumor as well as reduced VEGF expression compared to the tumors produced from wild-type cells [54]. Loss of VHL causes in constitutive HIF-1α activation and increases VEGF expression results in more hemorrhagic tumors with higher microvessel density in teratocarcinomas and fibrosarcomas compared to the tumors derived from wild-type cells [75,76]. Similarly, deletion of HIF-1α also leads to decreased VEGF expression and defective vascularization of tumors in nude mice [77]. HIF-1α also regulates the broad range of genes such as angiopoietins and VEGF receptors involved in the regulation of angiogenesis. Tang et al., reported that HIF-1α can down-regulate VEGFR-2 by regulating a VEGFR-1/VEGF/VEGFR-2 autocrine loop, which is essential for its post transcriptional induction in hypoxic conditions [78]. Loss of HIF-1α also disrupts this feedback mechanism, which is responsible for proper vasculogenesis, endothelial cell proliferation, tube formation, and growth of a solid tumor in vivo [78]. It was reported that endothelial cells cannot migrate properly to the hypoxic region due to the loss of HIF-1α [78]. Another study showed that HIF-1α regulates angiogenesis by controlling the VEGF/FLT1 signaling pathway in neuroblastoma cells [79]. FLT1, a “fms-like tyrosine kinase” receptor has a crucial role in regulating angiogenesis, migration of endothelial cells and cell survival in several types of cancer including prostate, colon, pancreas, glioblastoma, lymphoma, leukemia, and mesothelioma. FLT1 owns a binding site for HIF-1α. Moreover, VEGF/FLT1 activates HIF-1α via activation of ERK1/2 resulting in an alteration in VEGF level in tumor cells. This autocrine feedback loop is also associated with up-regulation of a potent anti-apoptotic protein, BCL-2 and an angiogenic factor bFGF indicating a significant role of HIF-1α in angiogenesis [79]. In addition, HIF-1α induces VEGF by recruitment of bone marrow-derived CD45+ cells which secretes metalloproteinase-9 (MMP-9) in the tumor site. A recent study reported that HIF-1α recruits bone marrow-derived CD45+ myeloid cells containing Tie2+, VEGFR1+, CD11b+, and F4/80+ subpopulations by inducing stromal-derived factor 1α (SDF1α) [80].

Moreover, several studies reported that the regulation of Nitric oxide synthases (NOS) is mediated by HIF-1α [81,82]. NOS isoforms produce nitric oxide (NO) which helps in survival of endothelial cell via inhibition of apoptosis induced by up-regulating caspase signaling [81,82]. NOS contains HRE site at its promoter. Quintero et al., reported that NOS is positively correlated with HIF-1α in squamous cell carcinoma [81]. In addition, inhibition of production of NO by the NOS inhibitor, l-NMMA prevents stabilization of HIF-1α in oral squamous carcinoma cell line [81]. HIF-1α also regulates angiogenesis via PI3K/AKT signaling pathway in human breast tumor and glioblastoma cells [83,84]. In addition, the PI3K/AKT signaling pathway encourages tube formation of HUVEC cells induced by bFGF activated by HIF-1α [85]. PDGF-A, PDGF-B, and EGF bind to the platelet-derived growth factor receptor (PDGFR) and epidermal growth factor receptor (EGFR), respectively. These bindings promote HIF-1α synthesis via the PI3K/AKT signaling cascade resulting in angiogenesis in cancer [86]. It is reported that activation of the EGFR/PI3K/AKT/mTOR pathway can increase VEGF expression by up-regulating HIF-1α [85]. Another study reported that PI3K/mTOR pathway increases HIF-1α protein levels without altering HIF-1α mRNA levels [85].

Moreover, Ravi et al., demonstrated that loss of p53 promotes angiogenesis in tumor xenografts in nude mice by HIF-1α regulation [71]. P53 helps in proteasomal degradation of HIF-1α mediated by Mdm-2 resulting in down-regulation of VEGF expression in colon carcinoma [71].

### 3.3. Role of HIF-1α in Metastasis

Malignant cells achieve a unique ability to disseminate from primary lesion to distal organ to form a secondary tumor. This phenomenon is called metastasis. This important hallmark of cancer is responsible for 90% of cancer-related lethality in patients [87]. Epithelial to mesenchymal transitions (EMT) is one of the most important characteristics gained by malignant cells to fulfill the goal metastasis. Loss of expression of epithelial marker proteins and gain of expression of mesenchymal marker proteins allow epithelial cells to detach from their neighboring cells by disrupting cell-cell attachment which helps cancer cell migration to distant sites. It is reported that hypoxia can stimulate metastasis by inducing EMT [88,89]. Yang et al., reported that Twist, an important mesenchymal marker, is directly regulated by HIF-1α [88]. Twist is a BHLH transcription factor and plays a pivotal role in regulation of transcription factors such as Snail in metastasis [88]. HIF-1α activates histone deacetylase 3 (HDAC3) which then binds to the promoters of CDH1 and junction plakoglobin (JUP) followed by transcription of Snail. HIF-1α promotes metastasis in cancer tissues via SMAD and non-SMAD signaling pathways by up-regulating TGF-β expression. Phosphorylated TGF-βRI binds with TGF-βRII, which in turn activates SMAD signaling pathway. This pathway regulation results in HIF-1α activation that causes the transcription of several EMT inducing genes. Non-SMAD signaling pathway is also triggered by HIF-1α via PI3K/AKT/mTOR signaling network. Along with this, WNT/β-catenin also plays a significant role in hypoxia-induced EMT by HIF-1α in malignant cells. In addition, HIF-1α activates the hedgehog signaling pathway, which helps in EMT of cancer cells [89]. HIF-1α is inversely correlated with the expression of E-cadherin in ovarian cancer cell lines, SKOV3 and OVCAR3 as well as in vivo showing an attachment between hypoxia and metastasis. In addition, regulation of E-cadherin expression is mediated by an up-regulation of Snail via HIF-1α in malignant cells [90]. Along with this, Lysyl oxidase (LOX) is also up-regulated in hypoxia. Erler et al., validated that HIF-1α regulates LOX, an extracellular matrix protein which helps in formation of premetastatic niche mediated by bone marrow cell recruitment in hypoxic conditions via a functional hypoxia-responsive element [91]. Their study reported the important role of HIF-1α in hypoxia-induced metastasis. Expression of LOX also has a significant correlation with hypoxia in breast cancer patients with ER-negative tumors and head and neck cancer patients [91]. HIF-1α also promotes EMT by regulating other genes involved in this process such as TCF3, ZEB1, and ZEB2 (Figure 2) [92]. In hypoxic conditions, ZEB1-MYB-E-cadherin signaling plays a key role in the activation of EMT. Moreover, HIF-1α also induces Jagged2, cyclooxygenase-2 (COX-2), and urokinase receptor (uPAR) which participate in metastasis via regulating EMT [15,16,17]. Not only via inducing EMT, HIF-1α also induces metastasis by regulating proteins involved in invasion. Cancer cells invade surrounding tissues by disrupting the basement membrane. Various reports suggest that Matrix metalloproteinases (MMPs) are involved in degradation of components of extracellular matrix. HIF-1α induces MMP-2 and MMP-9 expression which promotes invasion by degrading type IV collagen, an important component of the basement membrane (Figure 2) [92,93]. HIF-1α also induces expression of TGF-β in hypoxic conditions, which in turn activates its downstream regulators such as Smads, Snail, Slug, and Twist (Figure 2). On the other hand, HIF-1α also inhibits E-cadherin expression by restricting activity of its upstream regulator TGF-β [94,95]. To be metastatic, cancer cells have to extravasate into the tissue after its migration to distal organ where they have to proliferate again to form a secondary tumor. In hypoxic conditions, HIF-1α induces expression of receptor tyrosine kinase MET. MET is the specific receptor for Hepatocyte growth factor (HGF) which is a pleiotropic cytokine also known as scatter factor-1. This specific binding promotes invasion and extravasion in cancer tissues. However, this metastatic phenomenon is also accelerated through HIF-1α synthesis resulted by a positive feedback loop maintained by MET overexpression [96]. Additionally, HIF-1α has an important role in metastatic progression through up-regulation of NF-κB expression. Thus, accumulation of NF-κB causes overexpression of genes such as MMP-2/MMP-9 and activates urokinase-type plasminogen activator and chemokines such as stromal-derived factor-1alpha (SDF-1α). The receptors of SDF-1α, CXCR4 is involved in homing and migration of cancer cells [97,98]. Overexpression of inhibitor of *κB* (I*κB)* significantly inhibits CXCR4 expression resulting in inhibition of SDF-1α mediated cellular migration [99]. Moreover, NF-κB is also reported to repress E-cadherin and activate ZEB1 protein which are the downstream regulators of HIF-1α pathway indicating that NF-*κ*B and HIF-1α both depends on each other to induce EMT in tumor [100,101,102]. Therefore, HIF-1α not only has a prominent role in EMT, it can also regulate metastasis to accelerate the virulence of different cancers.

### 3.4. Role of HIF-1α in Cancer Stem Cell Proliferation and Maintenance

Cancer stem cells (CSCs) play an important role in cancer recurrence, metastasis, and therapy resistance. Low concentration of oxygen in cells or tissues, referred to as hypoxia, are one of the most invasive microenvironmental stresses. Hypoxia is the most common feature of solid tumors. Hypoxia is associated with many aspects of biological processes during tumor development and progression, such as cell survival, invasion, angiogenesis, and cellular metabolic alterations [19]. HIF-1α acts as a master transcription factor, can be stably expressed under hypoxia, and acts as a significant molecule to regulate the development of CSCs but the mechanism remains indistinct. Studies revealed that HIF-1α is related to the production of CSC markers [19]. The data indicates that HIF-1α can induce the production of multiple stem cell markers, such as OCT4, SOX2, Nanog, and Krüppel-like factor 4 (KLF4) (Figure 2) [19]. Additionally, the silencing of HIF-1α can hinder the progression of cancer by inhibiting the expression of stem cell markers. In the case of glioma, breast cancer, and prostate cancer, HIF-1α activates pro-survival pathways such as Notch, wingless, INT-1 (WNT), and the Hedgehog pathway, which are important for CSC maintenance, which leads to radioresistance and repopulate CSCs during or after treatment [103]. HIF-1α binds to the CD133 promoter and promotes the production of CD133^+^ glioma, colon, and pancreatic CSCs via OCT4 and SOX2. In turn, CD133 promotes HIF-1α expression and its translocation to the nucleus under hypoxic conditions. A recent study suggests that HIF-1α expression can be down-regulated by microRNA such as miR-935 in a feedback loop which in turn may inhibit glioma development [103]. There is a different opinion which is that expression of hypoxia-induced HIF-1α leads to a decrease in CD133 expression in gastrointestinal cancer cells that overexpress CD133. Under normoxic conditions, expression of HIF-1α is suppressed by the inhibition of mTOR signaling in CD133-overexpressing gastrointestinal cancer cells [104]. Under hypoxia while promoting differentiated cell phenotypes, HIF-2α silencing inhibits CSC phenotypes and is complementary to existing DNA alkylating treatments to inhibit glioma CSC activity. HIF-1α binds directly to the CD47 promoter to facilitate gene transcription, which helps to escape phagocytosis of macrophages and maintains the stem phenotype of breast CSCs. Endogenous HIF-1α promotes CD24 expression, as well as tumor formation and metastasis. In breast CSCs, the stability of Nanog mRNA through the transactivation of RNA demethylase ALKBH5 is increased by HIF-1α, and is involved in encoding N6-methyladenosine demethylase. 4-trimethylaminobutyraldehyde dehydrogenase (ALDH1A1), a subtype of aldehyde dehydrogenase, is associated with the self-renewal, metastasis, and resistance of cancer cells, is regulated by HIF-1α in breast cancer. Sequentially, ALDH1A1 promotes expression of HIF-1α via retinoic acid signaling. Zhang et al., reported the interrelation between CD47 and the cancer stem cell phenotype, but the molecular mechanisms of CD47 regulation have not been determined [105]. On the other hand, HIF-1α directly activates transcription of CD47 gene in hypoxic breast cancer cells. CD47 expression is enriched for cancer stem cells, and the depletion of cancer stem cell is led by deficiency of CD47. High CD47 expression is related to increased HIF-1α target gene expression. Thus, CD47 expression is fatal for breast cancer phenotype that is mediated by HIF-1α. The number of breast CSCs decreases upon inhibition of CD47 expression which is resulting in increased phagocytosis of breast cancer cell [105]. In prostate cancer samples, the co-localization of HIF-1α, OCT4 and Nanog suggest that the production of CSCs may be regulated by HIF-1α by regulating stem factors. In cervical cancer cells, OCT4B, an isoform of OCT4 promotes neovascularization by up-regulating HIF-1α production. In addition, HIF-1α also inhibits the expression of epithelial marker proteins, which can be confirmed by the use of HIF-1α inhibitors. Due to its association with neovascularization, HIF-1α can be used as a malignant marker of chondrosarcoma. Under hypoxic conditions, HIF-1α, a direct or indirect upstream regulator of the CSC marker proteins (Figure 2), may become a novel target to inhibit the signaling pathway of CSCs. In tumor progression, up-regulation of HIF-1α plays a vital role in CSC-regulated cancer hallmarks by controlling different gene expressions involved in CSC maintenance. Irradiation-induced DNA damage exerts intense regulation of HIF-1α, which not only depends on the oxygenation status and aberrant stabilization by Nijmegen breakage syndrome protein 1 (NBS1), but also induces EMT, invasion, and other characteristics of the CSC phenotype [106,107]. Due to localization of cancer stem cells (CSCs) in hypoxic niches, head and neck squamous cell carcinoma (HNSCC) is resistant to standard treatments. The NF-κB/HIF-1α signaling pathway maintains cancer stemness and high radioresistance in CD133-positive CSCs, whereas the inhibition of this HIF-1α involving pathway reverses the EMT and reduces the radioresistance in a model with laryngeal squamous carcinoma CNE-2 stem cells. Hypoxia-inducible factor-1α (HIF-1α) is involved in the resistance to photons, but its role in response to carbon ions remains unclear. HIF-1α mainly describes the radioresistance of CSCs of head and neck squamous cell carcinoma to both photon and carbon ion irradiation, which makes the HIF-1α targeting an attractive therapeutic challenge [108].

In high-grade serous ovarian cancer (HGSOC), a novel cancer stem cell (CSC) marker, ZIP4, while it converts to cisplatin (CDDP), it has been found that ZIP4 induced sensitization of HGSOC cells to histone deacetylase inhibitors (HDACis). On the other hand, ZIP4 selectively up-regulates HDAC IIa HDACs, with little or no effect on HDACs in other classes. With endothelial growth factor A (VEGFA), functional downstream mediators of HDAC4, and hypoxia-inducible factor-1 alpha (HIF-1α), HDAC4 knockdown (KD) and LMK-235 inhibit spheroid formation in vitro and tumorigenesis in vivo. Moreover, Fan et al., reported that ZIP4, HDAC4, and HIF-1α are involved in regulating secreted VEGFA in HGSOC cells [109]. While many HDAC4 targets have been identified, they focused on HIF-1α, one of the central players of tumor progression and drug response and VEGFA, one of the best-characterized HIF-1α targets. Acriflavine and linifanib, selective inhibitors for HIF-1α and VEGFA respectively, inhibit cell proliferation and block spheroid formation in both PE04 and PEA2 cells in HGSOC. In PEA2 cells, ZIP4-KD and HDAC4-KD reduce the level of HIF-1α. It is related to HDAC4’s effect on HIF-1α acetylation and stabilization. Additionally, ZIP4-KD and HDAC4-KD in PE04 and PEA2 extensively reduce VEGFA production/secretion in cell. LMK-235 and acriflavine considerably reduced VEGF production in PE04 and PEA2 cells [109].

Ammonia is a toxic by-product of metabolism that causes cellular stresses and it stabilizes and activates hypoxia-inducible factor-1α (HIF-1α). HIF-1α is also activated by ammonium chloride and compromises ammonia-induced apoptosis. Moreover, glutamine synthetase (GS), a key driver of cancer cell proliferation under ammonia stress and glutamine-dependent metabolism in ovarian cancer stem-like cells express CD90. Interestingly, activated HIF-1α counteracts the function of glutamine synthetase in glutamine metabolism by facilitating glycolysis and enriching glucose dependency. The functions of HIF-1α in a biphasic ammonia stress management in the cancer stem-like cells were unknown until now, where by GS facilitates cell propagation and HIF-1α contributes to the metabolic remodeling in energy fuel usage resulting in attenuated proliferation but conversely promoting cell survival [110]. In hypoxic conditions, some gastric CSCs improve the expression of hypoxia-inducible factor-1α (HIF-1α) and increase migration and invasion capabilities compared with the normoxic control. These CSCs are activated by mesenchymal cell marker Vimentin and by the inhibition of the epithelial cell marker E-cadherin. In this case, expressions of both HIF-1α and Snail increase, initiating a cascade of events that leads to the changes in characteristic of EMT, including decreased E-cadherin expression, increased Vimentin expression and enhanced invasion ability. Yang et al., reported that HIF-1α is responsible for activating EMT via increased expression of the transcription factor Snail in gastric CSCs [111]. Furthermore, the inhibition of Snail by shRNA reduces HIF-1α-induced EMT in gastric CSCs. As a result, hypoxia-induced EMT-like CSCs depend on HIF-1α to activate Snail, which may result in recurrence and metastasis of gastric cancer [111].

### 3.5. Role of HIF-1α in Cancer-Related Inflammatory Response

In the 19th century, Rudolf Virchow identified the presence of leucocyte within a tumor, indicating a role of inflammation in tumor progression and development [112]. Several studies reported that inflammation induced by many bacteria and viruses increases cancer risk [112,113]. As a result of this inflammatory response, various innate immune cells such as macrophages, neutrophils, mast cells, myeloid-derived suppressor cells, dendritic cells, and natural killer cells as well as adaptive immune cells such as T and B lymphocytes are recruited into tumor microenvironment. These cells communicate with each other using several autocrine and paracrine signaling methods mediated by cytokines and chemokines to regulate tumor growth [114]. HIF-1α plays a key role in generating an inflammatory response within the tumor microenvironment. Immune cells infiltrated at the tumor site also suffer from oxygen starvation resulting in activation of HIF-1α in them [57]. An important immune check point receptor, programmed death ligand 1 (PD-L1), is activated by HIF-1α resulting in myeloid-derived suppressor cells (MDSC)-mediated T cell activation [115]. Another important HIF-1α mediated inflammatory response occurs via activation of nuclear factor-kappa B (NF-κB) in tumor region. After dissociation from I-κB, NF-κB translocates into the nucleus and activates an array of proteins such as interleukin-6 (IL-6), cyclooxygenase 2 (COX-2), inducible nitric oxide synthase (NOS2), platelet endothelial cell adhesion molecule-1 (PECAM-1) and matrix metalloproteinase 9 (MMP9) [116,117]. Crosstalk between NF-κB and HIF-1α signaling cascades elevates inflammatory response in cancer by transcribing different downstream modulators such as IL-6, MMP9, and COX2 [118]. However, as reported by Uden et al., NF-κB binds to the promoter of HIF-1α at −197/188 bp from the initiation site and induces HIF-1α transcription [119]. Although HIF-1α is up-regulated by all the subunits of NF-κB, p50 and p52 show the highest and the lowest transcriptional efficacy [119]. In the case of translational regulation, NF-κB subunits such as RelA and c-Rel play the most important role in HIF-1α protein expression compared to other subunits such as p50, p52, and RelB [119]. Along with this, HIF-1α also regulates NF-κB activation in hypoxic conditions in vivo and in vitro in inflammatory conditions [120]. According to Han et al., HIF-1α alteration also occurs via TLR4 pathway which has a crucial role in inflammatory response at tumor site [121]. They proved that HIF-1α and its target gene VEGF become activated by the TLR activator, lipopolysaccharide (LPS) in HSC3 and SCC4 cells. Knockdown of TLR3 and TLR4 using siRNAs showed a significant reduction in Polyinosinic-polycytidylic acid (poly (I:C)) induced HIF-1α and VEGF mRNA expression [121]. They also showed that NF-κB up-regulates TNF-mediated HIF-1α and VEGF expression in oral squamous carcinoma cells. Moreover, inflammatory cytokines such as IL-1β, IL-6, IL-8, and IL-12p70 were also increased in hypoxic conditions, suggesting a role of HIF-1α in NF-κB mediated inflammatory response [121]. HIF-1α activates membrane receptors like RAGE and P2X7R which in turn induces NF-κB expression [97]. This inflammatory response includes chemokines such as CCL2, CCL5, and CXCR1/CXCL8, which have a crucial role in cell migration [122]. Ligands specific to RAGE and P2X7R receptors, such as HMGB1 and BzATP, are also induced inside the tumor due to the depletion of oxygen. This in turn helps in accumulation of NF-κB into tumor cells and exerts proinflammatory response [97].

## 4. Different Natural and Synthetic Compounds Targeting HIF-1α

Several studies revealed that HIF-1α could be a promising target for anticancer therapy. There are several phytochemicals and chemotherapeutic drugs leading in their anticancer effects by targeting HIF-1α and its related signaling pathways. Different types of cancers in the breast, colon, lung, prostate and ovary have regulation of HIF-1α; therefore, these cancers can be controlled by targeting this specific protein. Like other proteins, HIF-1α is synthesized into the ribosome and then it goes to its intended location for its specific activity and becomes degraded when its function passes. Therefore, the activity of HIF-1α may be inhibited by various means. It is possible to interfere with the mechanism of protein synthesis at the transcriptional or translational level. Other than these, the function of a protein can be impaired by inhibiting its DNA-binding or by proteasomal degradation of the protein directly. There is another aspect of HIF-1α inhibition by which one can restrict the activity of a protein indirectly by inhibiting its interacting subunits to be a potent inhibitor. Therefore, depending on the mode of action of HIF-1α targeting drugs, these could be classified in four major groups which include (i) HIF-1α synthesis blocker, (ii) HIF-1α activity blocker, (iii) HIF-1α degradation enhancer and (iv) Degrader of HIF-1α interacting HIF subunits. Based on the origin of the potent HIF-1α inhibitors, they can be broadly categorized in two major groups i.e., (a) natural compounds and (b) synthetic compounds. Here we are mentioning about the known promising inhibitors of HIF-1α which have potency to become useful antineoplastic therapeutics (Figure 3).

### 4.1. Natural Compounds as HIF-1α Inhibitors

There are several potential bioactive natural components exert their effects in different field of medicine including anticancer drug development. Various metabolites such as alkaloids, amines, alkamides, terpenes, steroids, saponins, flavonoids, tannins, phenylpropanoids, lignin, coumarins, lignans, polyacetylenes, fatty acids, and waxes are used for treatment of different type of cancers. Here, we discuss a few natural products which have an inhibitory role on HIF-1α and are able to regulate different cancer progressions (Table 1).

#### 4.1.1. HIF-1α Synthesis Blocker

First, we discuss the inhibition of de novo synthesis of HIF-1α by natural compounds. Silibinin (SL.1, Table 1) is a flavonolignan, found in fruit or seeds of milk thistle (*Silybum marianum*), has anti HIF-1α activity [123,124]. Silibinin is a clinically approved dietary compound against various liver diseases and cancers of the breast and prostate [125,126,127]. Dietary feeding of silibinin inhibits growth of advance human prostate carcinoma in athymic nude mice and increases plasma insulin-like growth factor-binding protein-3 levels [128,129,130,131]. Silibinin exerts its antitumor activity via inhibiting de novo synthesis of HIF-1α. Jung et al., suggested that Silibinin does not alter the transcription and degradation of HIF-1α, rather it inhibits translation of HIF-1α in LNCaP and PC-3 prostate cancer cells [132]. eIF4F complex is a key factor of HIF-1α translation which becomes altered by inhibition of phosphorylation of eIF-2a in Silibinin treated cells. Silibinin does not alter the mRNA level or half-life of HIF-1α rather it affects HIF-1α accumulation that promotes anticancer activity. A recent study by Deep et al., revealed that Silibinin inhibits HIF-1α synthesis by regulating its stimulator NOX [123]. In hypoxic conditions, NOX or NADPH oxidase generates ROS and promotes ROS mediated HIF-1α synthesis via PI3K/mTOR signaling pathways. Therefore, Silibinin has the potential to inhibit NOX mediated PI3K/mTOR signaling which in turn down-regulates HIF-1α synthesis [123]. Although Silibinin shows no significant toxicity, the prominent drawback of Silibinin is the inhibition of HIF-1α translation by it through targeting eIF4F complex which is a major factor of global protein synthesis. Considering the fact that eIF4F inhibition by Silibinin can cause normal cell death, researchers are interested to find other phytochemicals which can inhibit HIF-1α by exerting no toxicity or side effects.

In search of such effective bioactive components, researchers find DATS or Diallyl trisulfide (SL.2, Table 1), which is a water insoluble dietary organosulfur compound derived from the perennial plant *Allium sativum* (garlic). According to Wei et al., DATS is a natural Histone deacetylase that suppresses HIF-1α via inhibiting its upstream protein Trx-1 and attenuates metastasis of breast cancer [181]. However, DATS inhibits HIF-1α without promoting mRNA suppression or proteasomal degradation but it is able to inhibit HIF-1α synthesis at translational level [181]. Expression of L1CAM, VEGF-A, and EMT-related proteins (Slug, Snail, MMP-2, etc.) are repressed as a result of low HIF-1α level due to DATS treatment. Li et al., developed oil free microemulsion of DATS to reduce its toxicity and to improve its solubility and pharmacokinetics for making it a better anti HIF-1α compound for in vivo study [134]. Although these findings support DATS as a potent HIF-1α targeting anticancer agent, further investigation is needed into the detail mechanism of HIF-1α inhibition through Trx-1 suppression.

Other than these, Herboxidiene (GEX1A) (SL.3, Table 1) is an epoxide group containing polyketide molecule found to have anti HIF-1α activity. It is first isolated from the bacterium *Streptomyces chromofuscus* and known for its interference in the splicing of pre-mRNA of the gene that regulates cell cycle. According to Jung et al., in the case of HIF-1α, Herboxidiene inhibits splicing of it and decreases spliced HIF-1α mRNA level in hepatoma by targeting splicing factor 3B subunit 1 (SF3B1), the core spliceosome component [135]. Even though Herboxidiene intercedes in antitumor activity via inhibiting synthesis of HIF-1α, it has additive toxicity which needs to be reduced by further research.

Furthermore, Celastrol (tripterine) (SL.4, Table 1) is another phytochemical, isolated from the root extracts of *Tripterygium wilfordii* (Thunder god vine) and *Celastrus regelii*, used to improve apoptosis of cancer cells by targeting HIF-1α and inhibit HIF-1α mediated angiogenesis and metastasis. Growth, migration, and invasion of U87 and U251, human malignant glioblastoma cell lines, are inhibited by Celastrol [136]. Under hypoxic conditions, Celastrol inhibits HIF-1α mRNA levels and the hypoxia-induced accumulation of nuclear HIF-1α protein levels which subsequently resulted in the reduction of the transcriptional activities of HIF-1 target genes including VEGF. Celastrol also down-regulates the activities of PI3K, Akt, and mTOR signaling pathways which are the prominent regulators of HIF-1α synthesis. The expression of HIF-1 and vascular endothelial growth factor (VEGF) is promoted by LMP1 dependent JNK activation in Epstein-Barr virus (EBV)-associated nasopharyngeal carcinoma (NPC), which ultimately contributed toward radio-resistance in NPC patients. Phosphorylation of p38 MAPK and JNK1/2 in Cisplatin-resistant human NPC-039 and NPC-BM cells is increased by the treatment with Celastrol, increasing cytotoxicity by activation of caspase-mediated apoptotic pathways in these cells. The activation of JNK and p38 MAPK by LMP1 is related to the development of radio-resistance in NPC, but the mechanism remains to be unstated. A platelet-derived endothelial cell growth factor thymidine phosphorylase (TP) is related with poor prognosis in EBV associated NPC and expression of TP can be induced by triggering p38 MAPK pathway via the CTAR1 and CTAR2 domains of LMP1. In addition, anti-metastatic protein tissue inhibitor of metalloproteinase-3 (TIMP-3) through transcriptional repression via the p38 MAPK pathway is inhibited by LMP1 to promote metastasis [137]. As JNK and p38 MAPK pathway activation regulates HIF-1α expression, inhibition of these pathways by Celastrol may infer a negative effect on HIF-1α mediated cancer progression. To overcome the limitations of Celastrol, different derivatives of it were generated by Michael addition and ring-opening polymerization, which helps to make a compound with better solubility and pharmacological activities [138].

2-Phenethyl isothiocyanate (PEITC) (SL.5, Table 1) is another natural dietary HIF-1α synthesis blocker which is reported to have inhibitory effects on expression of HIF-1α target genes such as CAIX, GLUT1, BNIP3, and VEGF-A. PEITC does not depend on the activity of PHD and VHL, rather it inhibits phosphorylation of translational regulator 4E-BP1 resulting in attenuation of HIF-1α protein translation [139,140]. PEITC entered clinical trials for leukemia, whereas a phase II clinical trial has been completed for lung cancer [127]. PEITC has already completed phase I & II trials as a dietary supplement in oral cancer with mutant p53 [133]. However, it will be fascinating if PEITC could be developed as an anticancer drug by improving its narrow therapeutic window against a few types of cancer.

#### 4.1.2. HIF-1α Activity Blocker

Other than inhibition of HIF-1α synthesis, its activity could also be regulated by transcriptional inhibition of HIF-1α target genes. There are several compounds which inhibit HIF-1α directly by interfering its binding to the DNA or its cofactor. After hypoxia, HIF-1α binds to its other subunits and forms HIF complex. There are some coactivators of HIF-1α like CBP/p300 helping transactivation of the genes. Some HIF-1α inhibitors target the binding of HIF-1α to the HRE element or they may hinder interaction of HIF-1α and CBP/p300.

Echinomycin (SL.6, Table 1) is such a cyclic peptide from the quinoxaline antibiotic family that inhibits HIF-1α activity by restricting HIF-1α-HRE or HIF-1α-CBP/p300 interaction. It is extracted from a bacterium *Streptomyces echinatus* [141]. Chromatin immunoprecipitation assay by Kong et al., showed that Echinomycin specifically hampers the HIF-1α binding to the promoter of VEGF in U251, human glioma and MCF-7, breast cancer cell lines. The main recognition sites of Echinomycin are 5′-ACGT-3′ and 5′-TCGT-3′ of VEGF promoter which possess common central 2-bp sequence 5′-CG-3′ that serves as strong binding sequence [142]. Echinomycin was introduced in phase I–II clinical trials for B16 melanoma and the P388 leukemia, but it was rejected because of side effects and low antitumor effects. A recent study by Vlaminck et al., explained the dual role of Echinomycin on HIF-1α activity [143]. They proposed that under normoxic conditions, it increases HIF-1α level. According to them, this drug is not efficient enough as an antitumor agent down-regulating HIF-1α. In the recent studies by Bailey et al., Echinomycin is used as liposomal nanoparticle against HIF-1α in triple negative breast cancer to improve its efficacy [144]. They proposed that liposomal Echinomycin shows more therapeutic efficacy and less toxicity than cremophor formulated Echinomycin. In another study, Wang et al., proved the therapeutic effects of Echinomycin by targeting HIF-1α in TP53 mutated acute myeloid leukemia (AML) [145]. TP53 mutation occurs in AML with 10% frequency and associated with poor prognosis. According to Wang et al., PEGylated liposomal Echinomycin prolongs the drug circulation time in the bloodstream [145]. Additionally, they showed that Echinomycin not only inhibits AML cell metastasis but also it targets CD34+CD38- stem cell population by inhibiting HIF-1α. These findings claim the reformulation and reintroduction of Echinomycin in clinical trial for cancer therapy.

Looking for an alternative, Min et al., discussed the anticancer role of Chetomin, (SL.7, Table 1), an antimicrobial metabolite extracted from the mesophilic saprophytic fungus *Chaetomium globosum* [146]. They reported the Chetomin-attenuated sphere forming ability of CSC and proliferation ability of non-CSC in Non-small cell lung cancer (NSCLC) [147]. The Co-Immunoprecipitation analysis by Min et al., demonstrated the interaction between the HIF-1α and HSP90 proteins in hypoxia mediated-activation of HIF-1α [147]. HSP90 (Heat shock protein 90) is a chaperone, frequently overexpressed in some cancers leading to chemoresistance and lower survivability of patients. Binding to the PAS-B domain of HIF-1α, HSP 90 stabilizes it and helps in the activity of HIFs [148,149]. Chetomin significantly affects the binding between HSP90 and HIF-1α without affecting the level of HSP90 or HSP70. They tested the inhibitory role of Chetomin on adhered monolayer cultures and spheroid cultures in a dose-dependent manner. Chetomin exposure inhibits survival promoting factors such as mitogen-activated protein kinase 1/2 (MEK1/2), insulin-like growth factor 1 (IGF1 R), epidermal growth factor receptor (EGFR), SRC, activation of protein kinase B (AKT) and mammalian target of rapamycin (mTOR), basic fibroblast growth factor (bFGF), platelet-derived growth factor (PDGF), etc. in monolayer cultures. Immunohistochemical analysis of lung revealed lower intensity of HIF-1 α and CD34 in Chetomin treated group. Thus, Chetomin alters the CSC mediated chemoresistance via attenuation of the activity of HIF-1α. In another study by Kung et al., it is shown that Chetomin constrains HIF-1α activity, precluding the binding of HIF-1α to p300 [150]. HIF-1α interacts with p300 through CH1(cysteine/histidine-rich 1) domain of p300. The CH1 domain is a zinc-containing transcription adapter zinc-binding (TAZ) domain. Chetomin disrupts this CH1 domain and prevents the interaction between HIF-1α and its coactivator p300. From these observations, it can be inferred that Chetomin inhibits HIF-1α directly and it can be a potent inhibitor of HIF-1α.

Apart from these, Gliotoxin (SL.8, Table 1) is a sulfur containing mycotoxin which mediates same kind of activity such as Chetomin. It is extracted from the marine Trichoderma species *Gliocladium fimbriatum*. It is the most effective member of epidithiodioxopiperazine fungal toxin family and exerts its anti-HIF-1α activity, down-regulating the transcription of VEGF-A, LDHA, and ENO1 genes. Reece et al., studied the antitumor activity of Gliotoxin in prostate cancer and proposed that it can disrupt the HIF-1α/p300 complex in vitro and in vivo [151]. Using a fluorescence binding assay, they confirmed that Gliotoxin disrupts the CTAD domain of HIF-1α and the CH1 domain of p300 which in turn interrupts their binding. Despite having such negative effects against HIF-1α, it has not started clinical trials because of its high toxicity. In earlier studies, it was observed that Gliotoxin treatment leads to death of experimental mice [152]. Therefore, further studies are going to need to reduce the toxic effects of Gliotoxin. In this context, Hubmann et al., proposed that lower dose of Gliotoxin shows no significant toxicity [153]. However, Comas et al., delivered Gliotoxin in cancer cell via nanoparticle and showed its antitumor activity with low toxicity [154]. The effects of Gliotoxin are also studied in combination with other chemotherapeutic agents in very low dose for reducing its toxic effects.

However, to overcome such toxic effects of therapeutics and to establish a new effective drug, Xia et al., introduced Sulforaphane (SL.9, Table 1), which is a nitrogen containing isothiocyanate compound that can be found in cruciferous vegetables such as broccoli, kale, cabbage, and watercress [155]. Xia et al., enlightened the anti HIF-1α activity of Sulforaphane in the case of non-muscle invasive bladder cancer cell line [155]. They explained that Sulforaphane blocks translocalization of HIF-1α to nucleus and suppresses hypoxia mediated glycolysis. In another study by Kim et al., Sulforaphane is introduced as an anti-HIF-1α component against human colon cancer and gastric cancer cells [156]. At first, Kim and his colleagues thought that inhibition of HIF-1α by Sulforaphane may be due to degradation by 26S proteasomal pathway. Further studies by Kim et al., confirmed that inhibitory effect of Sulforaphane against HIF-1α is neither 26S proteasomal pathway dependent nor Akt/mTOR signal mediated, rather they observed lysosomal degradation of HIF-1α [156]. Sulforaphane promotes lysosomal activity and degrades HIF-1α in hypoxia. There is no significant toxicity in Sulforaphane treatment and it is easily metabolized via the mercapturic acid pathway [156]. Although Sulforaphane undergoes phase IV clinical trials for treating Human immune deficiency virus (HIV) infection and phase II trial sfor combating autistic disorders, neurodevelopmental disorders, diabetes mellitus, and schizoaffective disorders, in the case of cancer there are only a few reports which mention phase II clinical trials for adenocarcinoma and recurrent prostate cancer [127,133]. Therefore, Sulforaphane could be inspected further as an anti-HIF-1α molecule.

One of the most potent inhibitors of HIF-1α activity is Acriflavin (ACF) (SL.10, Table 1). It is extracted from coal tar and was first introduced in medical science in 1972 by the German researcher Paul Ehrlich. ACF was generally used as antiseptic, trypanocides, anti-viral, and anti-bacterial agent. In recent years, ACF is repurposed in cancer to develop new therapeutic strategies [157]. Nehme et al., and Cheloni et al., demonstrated the anti-oncogenic properties of ACF against chronic myeloid leukemia [158,159]. Even though ACF is an FDA approved drug for urinary tract infections, recent research proved its anti-cancer efficacy which involves inhibition of HIF-1α activity [127,133,159]. Mangraviti et al., demonstrated that ACF down-regulates proangiogenic protein VEGF, PGK-1 by inhibiting the formation of HIF-1α and HIF-1β heterodimer in brain cancer [160]. According to Mangraviti and his colleagues, ACF can inhibit transcriptional activity of HIF-1α not only in brain cancer but also in cholangiocarcinoma (SK-ChA-1), ovarian (A2780), or breast cancer (MCF-7) cell lines [160]. They also explained that ACF has cytotoxic effects on glioma stem cells (GSCs) and reduces chemoresistance. ACF inhibits HIF-1α through binding to the PAS-B domain which in turn restricts its binding to HIF-1β subunit necessary for its transcriptional activity. The expression of hypoxia-induced GLUT1, a typical HIF-1α target gene, is reduced by the treatment with ACF [161]. Therefore, ACF could be used individually or in combination with other drugs as an antiangiogenic agent or a CSC inhibitor in cancer treatment.

In addition, Huang et al., proposed another phytochemical Emodin (SL.11, Table 1) as a HIF-1α inhibitor, enhancing antineoplastic effects of other chemotherapeutic drugs [162]. Emodin is an anthraquinone derivative extracted from the Chinese herbs such as *Rheum palmatum* and *Polygonam multiflorum*. Cancer cells explore multiple drug resistance against chemotherapeutic drug by MDR1 gene. MDR1 is a downstream targeted gene of HIF-1α. Generally, in cancer cells, ROS and HIF-1α maintain a balance. Increasing the ROS concentration causes the activation of HIF-1α and other members of the HIF family [163]. As a result, MDR1 gene becomes transcriptionally activated by HIF-1α and produces multidrug resistance. On the other hand, Huang and his colleagues showed that in combination therapy of Emodin with Cisplatin, ROS becomes increasingly relative to control and exerts ROS mediated HIF-1α suppression. By in vitro and in vivo study of Emodin cotreatment, it is documented that multiple drug resistance becomes altered as a result of the down-regulated MDR gene. As Emodin is a ROS generator, it suppresses transactivation of HIF-1α in combination therapy with Cisplatin to DU-145 cells [162]. Despite the fact that Emodin has failed to prove its clinical efficacy against polycystic kidney disease, its significant anti-neoplastic effects are encouraging researchers to include it in the preclinical study list of anticancer therapeutics.

Other than these phytochemicals, the lactone ring containing group of steroids, Cardenolides (SL.12, Table 1) also show attenuation of HIF-1α transcriptional activity. The latex containing Cardenolides are extracted from various medicinal plants belonging to the family *Asclepidaceae* and *Apocynaceae*, native to India, Africa, southern China, and southeast Asia. Parhira et al., isolated cardenolides from *Calotropis gigantea* and demonstrated their anticancer activity [164]. They isolated 20 Cardenolides, among which Digoxin has started clinical trials for breast cancer, colon cancer, rectal cancer, and sarcoma therapy. Although Digoxin has either completed or started phase IV clinical trials for arterial fibrillation, cardiac failure, Dilated Cardiomyopathy, and acute kidney injuries, the addition of these compounds to clinical trials is comparatively new [127,133]. Zheng et al., isolated six new non-classical Cardenolides, including 17 known ones [165]. 19-dihydrocalotoxinis one of those 23 compounds, which has more potential as an anti-HIF-1α component compared to Digotoxin. Further studies are necessary to verify its toxic effects and anti-HIF-1α activity in different cancers.

#### 4.1.3. HIF-1α Degradation Enhancer

Degradation of an oncogenic protein by drug treatment could be a promising method for anticancer therapy. There are several HIF-1α inhibitors which can act by proteasomal degradation of the protein; thus, they represent a different approach compared with the other two mechanisms of HIF-1α inhibition stated earlier. 3,3′-Diindolylmethane or DIM (SL.13, Table 1) is one of those compounds, which exerts its anticancer effect by executing proteasomal degradation of HIF-1α. DIM is a phytochemical derived from Indole-3-carbinol (I3C), present in cruciferous vegetables [166]. Riby et al., demonstrated the inhibitory effects of DIM on HIF-1α in a dose-dependent manner [166]. They explained that DIM has a dual role in both synthesis and degradation of HIF-1α in MDAMB-231 cell line. To assure the effects of DIM on synthesis of HIF-1α they inhibited proteasomal degradation of HIF-1α using MG132, a proteasome inhibitor, and measured accumulation of HIF-1α protein level which became significantly reduced after DIM treatment. Furthermore, the rate of HIF-1α degradation was halted at a certain concentration of DIM which suggests the saturation of its activity. From this study, it was concluded that DIM significantly reduces the synthesis of HIF-1α and it mediates the degradation of HIF-1α with a concentration independent mechanism. Furin and VEGF are two HIF-1α regulated proangiogenic endogenous genes which are inhibited by DIM treatment in hypoxic tumor cells in concentration-dependent manner. Due to proven preclinical efficiency of DIM, it started different phases of clinical trials against different cancers such as cervical and prostate cancer [127,133]. As DIM is a natural compound and it was previously proved that DIM has no side effects in patients and it has a profound effect on HIF-1α, it could be one of the potent therapeutics against cancer targeting HIF-1α [167,168].

Additional studies of Li et al., displayed antiangiogenic role of pseudolaric acid B(PAB) (SL.14, Table 1) via proteasomal degradation of HIF-1α in MDAMB-468 cell line [169]. PAB is a natural diterpenoid antifungal compound occurring from *Pseudolarix kaempferi*. Li et al., stated that PAB inhibits paracrine secretion of VEGF via lowering the level of HIF-1α [169]. However, transcription of HIF-1α is not affected by PAB. On the contrary, HIF-1α level becomes decreased via ubiquitin proteasomal degradation mechanism. PAB has a dual effect as an antiangiogenic agent, i.e., not only has it an inhibitory effect on HIF-1α but also has a negative effect on the endothelial cell directly. PAB can also indirectly target HIF-1α stabilization by targeting c-Jun, cellular proliferation marker [170]. PAB phosphorylates c-Jun at Ser63/73 which causes destabilization of HIF-1α protein. This activity of PAB leads to premature degradation of HIF-1α protein and transcriptional attenuation of VEGF and MDR1 genes. For these antiangiogenic effects of PAB, it can be novel anticancer therapeutics for clinical studies.

Bavachinin (SL.15, Table 1) is another HIF-1α degrader and a natural prenylated flavanone, present in the Chinese herb *Psoralea corylifolia*. It has potent anti-angiogenic activity and an anti-inflammatory function. Under hypoxic conditions, Bavachinin decreases the activity of HIF-1α in a concentration-dependent manner in human KB carcinoma and HOS osteosarcoma cells, mainly by raising the interaction between VHL and HIF-1α followed by proteasomal degradation. Moreover, the downstream genes of HIF-1α (e.g., VEGF, GLUT-1, and hexokinase-2), which block angiogenesis and reduce energy metabolism, are suppressed by Bavachinin. Additionally, Bavachinin prohibits tube formation by HUVECs and also prevents the migration of KB cells. In KB cells, Bavachinin reduces the expression of CD31 [171]. As there is no significant toxicity is shown in Bavachinin treatment, it can be further investigated.

Aside from these, Ma et al., showed that Andrographolide (AGL) (SL. 16, Table 1) promotes proteasomal degradation of HIF-1α and hinders the nuclear localization of it in Hep3B and HepG2 liver cancer cell lines [136]. AGL is a bioactive diterpenoid isolated from *Andrographis paniculate* belonging to family *Acanthaceae* [136,172]. Inhibition of nuclear localization of HIF-1 α by AGL leads to transcriptional down-regulation of HIF-1α targeted genes. Cui et al., reported that the AGL is a HIF-1α and VEGF inhibitor targeting the PI3K/AKT signaling pathway and preventing choroidal neovasculature [173]. Therefore, despite having efficacy against acute exacerbation of chronic bronchitis and acute tonsillitis, these anti-oncogenic preclinical data support repurposing of AGL against different cancers [127,133]. Although the compound is in a phase III clinical study for squamous cell carcinoma, more research needs to be carried out to improve the activity of AGL [127,133].

Isoliquiritigenin or ILTG (SL.17,Table 1) facilitates its anticancer activity via the proteasomal degradation of HIF-1α and inhibits kinase activity of VEGFR-2(VEGF receptor 2) [174]. It is a chalcone type dietary flavonoid turned out from the roots of licorice plants such as *Glycyrrhiza uralensis*, *Mongolian glycyrrhiza*, and *Glycyrrhiza glabra* [175]. In vitro and in vivo study of Wang et al., validated that ILTG treatment inhibits cancer neo-angiogenesis in breast cancer and showed no significant toxicity. However, establishment of ILTG as anticancer therapeutics claims needs more research [174].

Wondonin (SL.18, Table 1) as a HIF-1α degrader, has significant potential in clinical applications against cancer. It is a bis-imidazole compound isolated from the sponge, *Poecillastra wondoensis*. Jun et al., proposed that Wondonin down-regulates the immunoreactivity of CD31 and VEGF, whereas it increases the affinity of binding of pVHL to HIF-1α, which leads to proteasomal degradation of HIF-1α [176]. Therefore, Wondonin has a prominent antiangiogenic role as it controls both HIF-1α and VEGF. Thus, Wondonin can be a novel anticancer therapeutic agent against HIF-1α.

Lee et al., introduced Thymoquinone (SL.19, Table 1) as another HIF-1α targeting antineoplastic benzoquinone phytochemical isolated from the seed of black cumin (*Nigella sativa*) [177]. Thymoquinone is a popular compound in phase II or phase III clinical trials for COVID-19 and polycystic ovarian syndrome [127,133]. In the case of cancer, researchers conducted phase II clinical trials of Thymoquinone against premalignant lesion [127,133]. However, according to the preclinical studies, Thymoquinone destabilizes the binding between HIF-1α and HSP90 and helps in pVHL-mediated polyubiquitination of HIF-1α protein, which is independent of prolyl hydroxylation in renal cancer cells [177]. Recent research by Homayoonfal et al., showed that targeting microRNAs with thymoquinone could be a new approach for cancer therapy [182]. PEGylated Thymoquinone up-regulates expression of miR-361 which suppresses proinflammatory and proangiogenic genes including HIF-1α. Thus, Thymoquinone could be a potent HIF-1α inhibitor which enhances proteasomal degradation of HIF-1α. Hence, it is an intriguing field of research where more drugs degrading the oncogenic HIF-1α are yet to be discovered.

#### 4.1.4. Degrader of HIF-1α Interacting HIF Subunits

Earlier in this review, we discussed the natural compounds that target HIF-1α by blocking its synthesis, inhibiting its activity, and degrading the protein. Here another mechanism of HIF-1α inhibition is discussed, which is about down-regulation of its interacting subunits. As discussed in this review, HIF-2α and ARNT are the key interacting subunits of HIF-1α, expressing changes which can be noticed in hypoxia.

Strofer et al., suggested that at higher Curcumin (SL.20, Table 1) concentration, HIF-1α, HIF-2α and ARNT level decline in Hep3B, HepG2, and MCF-7 cancer cell lines [179]. Curcumin ((1,7-bis(4-hydroxy-3-methoxyphenyl)-1,6-heptadiene-3,5-dione) is a polyphenol compound extracted from rhizomatous herbaceous plant *Curcuma longa* (turmeric). For thousands of years, turmeric has been renowned for its medicinal value. At present, it is considered to be one of the FDA approved drugs in autosomal dominant polycystic kidney disease (ADPKD), chronic schizophrenia, major depressive disorder, periodontitis and type-2 diabetes mellitus [127,133]. Curcumin has been introduced into phase II and III clinical trials against various cancers such as breast, colorectal, and head and neck cancers [127,133]. Prolyl hydroxylase is an enzyme that hydroxylates HIF-1α and helps in proteasomal degradation in the presence of oxygen. Oxidation of iron is required for this oxygen mediated regulation of HIF-1α protein degradation. Curcumin acts as an iron chelator and disrupts enzymatic activity of the enzyme prolyl hydroxylase. Thus, Curcumin has miscellaneous effects on HIF-1α regulation [179]. In another study, the inhibitory role of Curcumin against HIFs was studied by Sarighieh et al., on MCF-7 cells and cancer stem- cells (CSCs) [180]. Presently, researchers focus on therapeutic efficacy of different synthetic drugs and natural compounds against CSCs (cancer stem cells). Sarighieh and his colleagues studied the role of curcumin on HIF-1α and HIF-2α in a dose-dependent manner. They revealed that Curcumin could not inhibit HIF-1α directly, rather it mediates antiangiogenic and antiproliferative effect via inhibiting HIF-2α and ARNT in both normoxic and hypoxic conditions, whereas HIF-1α level was not affected. Therefore, Curcumin interferes with metastasis by indirectly targeting HIF-1α by degrading HIF-2α and ARNT in CSCs of MCF7 and MDAMB-231 cell line [180]. Although, Curcumin is a well-known and effective natural compound with no side effects, it also has negative effects on CSCs, the momentous element of cancer cell metastasis, drug resistance and recurrence. Therefore, Curcumin could be a coercive element against HIF-1α mediated CSC progression.

The aforementioned natural compounds not only have fewer side effects, but they can also possess alternative anti-oncogenic response which involves targeting inflammatory marker such as NF-κB, a prominent pathway modulator of HIF-1α. Out of these natural compounds, Emodin has the ability to reduce the expression of NF-κB, whereas most of the natural compounds such as Silibinin, Celastrol, PEITC, DIM, Andrographolide, Thymoquinone and Curcumin can act as a inhibitor of IκBα degradation, which in turn cause suppression of phosphorylation, acylation, and nuclear translocation of NF-κB [178,183,184,185,186,187,188,189,190,191,192]. On the other hand, Gliotoxin is a natural compound that inhibits NF-κB activity by up-regulation of ROS, whereas, Sulforaphane decreases NF-κB activity by preventing the interaction between NF-κB and its consensus sequence [193,194].

Further research is going on for proving anticancer as well as anti-HIF-1α efficacy of different nontoxic and novel phytochemicals which need to be scrutinized for their better functionality.

### 4.2. Synthetic Compounds as HIF-1α Inhibitors

Conventional chemotherapy mostly includes synthetic drugs which have a specific effect on different cancers. Although synthetic drugs possess cytotoxic effects, they can interfere with cancer cell division, metastasis, proliferation, etc. Therefore, targeting oncogenic protein-like HIF-1α with synthetic drugs with low toxicity is a fascinating part of anticancer drug discovery. The common chemotherapeutic synthetic drugs able to regulate HIF-1α are listed in Table 2.

#### 4.2.1. HIF-1α Synthesis Blocker

Some synthetic compounds also act as HIF-1α synthesis blockers that have different modes of action. 1-benzyl-3-(5′-hydroxymethyl-2′-furyl)-indazole or YC-1 (SL.1, Table 2) is such a compound which is introduced as HIF-1α inhibitor by Chun et al. [195]. Earlier studies demonstrated that YC-1 is a soluble guanylate cyclase inducer used in cardiopulmonary disease [196]. From the current research of Sun et al., it was proved that YC-1 can exert anticancer effect by targeting HIF-1α expression and activity through PI3K/AKT/mTOR/4E-BP axis regulation [197]. A recent study reported that YC-1 is able to block the induction of erythropoietin (EPO) and VEGF mRNA by inhibiting HIF-1α during hypoxia [198]. Along with these, several inflammation markers such as IL-6 and IL-8 are also significantly decreased by YC-1 treatment in vivo [199]. YC1 displayed excellent anti-cancer effects in different cancer cells including lung cancer, bladder cancer, breast cancer, canine lymphoma, colorectal cancer, gastric carcinoma, hepatocellular carcinoma cancer (HCC), preeclampsia, ovarian cancer, and non-small cell lung cancer (NSCLC) [200]. Another study stated that YC-1 can reverse the acquired resistance to gefitinib by inhibiting HIF-1α in HCC827 GR cells [201]. Additionally, Li et al., proposed that YC-1 exerts FIH dependent inhibition of p300 recruitment to HIF-1α and inhibits hypoxia mediated activation of NFκB, a downstream target of HIF-1α [202]. As NFκB is a key regulator of CSC maintenance and cancer cell proliferation, YC-1 could be a novel anticancer drug which may target cancer stem cell propagation as well as several other hallmarks of cancer such as angiogenesis and metastasis.

Another example of an HIF-1α inhibitor is PX-478 (S-2-amino-3- [40-N, N-bis(2-chloroethyl) amino] phenyl propionic acid N-oxide dihydrochloride) (SL.2, Table 2). PX-478 started clinical trials against lymphoma and other advanced metastatic cancer. VHL mediated ubiquitination has no role in PX-478 dependent down-regulation of HIF-1α expression, rather it was observed that it can cause a reduction of the HIF-1α m-RNA level. PX-478 also affects transcription of HIF-1α without altering eIF-4E/4E-BP axis [203]. According to Welsh et al., the transcription of VEGF and GLUT-1 are decreased in vitro and in vivo after PX-478 treatment in colon cancer [203].

Furthermore, PX-478 helps in radiosensitization of hypoxic C6 glioma, HN5, and UMSCCa10 squamous cells and Panc-1 pancreatic adenocarcinoma cells by inhibiting HIF-1α in vitro. Along with these, PX-478 showed excellent radiosensitization of tumors by down-regulating post-radiation HIF-1α signaling pathways. PX-478 showed promising anti-cancer effects in phase I clinical trials with 41 patients at two sites in the USA. PX-478 is reported to be a well-tolerable drug in a relatively high proportion of patients. These pre-clinical and clinical studies decipher that PX-478 could be a potent anti-cancer agent by targeting HIF-1α in the near future [204].

#### 4.2.2. HIF-1α Activity Blocker

Generally, the synthetic chemotherapeutic HIF-1α inhibitors may have specific and unique mechanism. For the development of an HIF-1α targeting cancer therapy, pathway-based regulation of HIF-1α with DNA-binding small molecules may represent an important approach.

Cisplatin is such a common synthetic anticancer agent that can down-regulate the expression of HIF-1 α in the Cisplatin-sensitive cell line by inducing apoptosis via elevating the expression of the cleaved PARP in dose and time dependent manner. Ai et al., tested the inhibitory role of Cisplatin against HIF-1α with two pairs of genetically matched Cisplatin-sensitive and Cisplatin-resistant ovarian cancer cell lines [223]. However, in normal conditions, the Cisplatin-resistant cell line showed inhibition in apoptosis, which was altered with low expression of HIF-1α. As discussed earlier, HIF-1α can regulate CSC properties that are responsible for drug resistance and cancer recurrence. For the researchers it is intriguing to find therapeutic modalities that can overcome HIF-1α mediated CSC dependent cancer resistance and recurrence. In this context, Zhihong and his colleagues showed that cancer recurrence and metastasis occurred in patient after Cisplatin treatment and PEO4 cells were derived from that patient after acquiring Cisplatin resistance, whereas PEO1 is the Cisplatin-sensitive primary ovarian cancer cells from the same patient before Cisplatin treatment [223]. A2780/CP is a Cisplatin-resistant ovarian cancer cell line which is derived from A2780, Cisplatin-sensitive ovarian cancer cell line., HIF-1α is down-regulated in A2780 and PEO1 cell line after Cisplatin treatment and the cancer cells undergo apoptosis, whereas Cisplatin sensitivity restores after HIF-1α knockdown and induces apoptosis in Cisplatin-resistant cell line A2780/CP and PEO4. From this study, it is confirmed that HIF-1α is an important factor in drug resistance and it can be targeted by Cisplatin [223].

Along with these, we focus on some other synthetic chemotherapeutic drugs which inhibit HIF-1α directly by interfering its binding to DNA. Jones and Harris explored the role of three suppositional molecules DJ12 (SL.3, Table 2), DJ15, and DJ30 against HIF-1α where DJ12 is more effective [205]. The effect of DJ15 and DJ30 is cell line specific. They studied the role of DJ12 in breast cancer cell lines MDA-MB-468 and ZR-75, melanoma cell line MDA-MB-435, and pVHL mutant renal cancer cell lines RCC4 and 786-O and concluded that DJ12 not only blocks the binding of HIF-1α to HRE region of DNA but also interferes with its transactivation. This leads to a reduction of mRNA expression of VEGF and BNIP3. With a HRE binding assay of nuclear and cytoplasmic extract of MDA-MB-468 cell line, it is confirmed that DJ12 did not affect DNA protein-binding directly, rather it inhibited the formation of HIF-1α, HIF-1β, and CBP/p300 transcription complex or folding of HIF-1α. Therefore, DJ12 can be used as a significant HIF-1α inhibitor in various cancer types.

Furthermore, Sung et al., studied the role of another synthetic drug Bortezomib (PS-341) (SL.4, Table 2) against HIF-1α in Hep3B (human hepatoma), ARH77 (human multiple myeloma), and U299 (human multiple myeloma) cell lines [206]. Bortezomib is a clinically approved proteasome inhibitor for myeloma and different solid tumors. Bortezomib mediates its effect on HIF-1α via interacting with CAD (C -terminal transactivation domain) and inhibits recruitment of p300 coactivator. In some cases, inhibition of HIF-1α by Bortezomib is improved by FIH (factor inhibiting HIF-1) hydroxylation. Bortezomib helps in prolyl hydroxylation at Pro402 and Pro564 by PHD (HIF-1 prolyl hydroxylases) and hydroxylation at Asn803 by FIH in hypoxic conditions [207]. These activities lead to repression of transcriptional activity of HIF-1α and inhibition of erythropoietin (EPO), vascular endothelial growth factor (VEGF), and carbonic anhydrase IX (CAIX) genes, downstream target of HIF-1α. Bortezomib is introduced to 15 clinical trials as monotherapy and 17 clinical trials as combinatorial therapy with other drugs. Bortezomib monotherapy showed poor and minimal effects in clinical studies. These results reported that Bortezomib offered no hope as a single agent as anti-cancer drug. Other clinical trials reported that Bortezomib with other drugs showed fewer anti-cancer activities with no statistical significance. In a phase I/II dose escalation study, it was observed that Bortezomib has anti-cancer effects and it has been tolerated by Advanced Androgen-Independent Prostate Cancer patients when combined with Docetaxel. However, further clinical trials using combinatorial therapy of Bortezomib with other drugs are needed for better understanding of its anti-cancer effects [208].

Additionally, several antimicrobial and antifungal agents play an antiangiogenic role targeting HIF-1α. Amphotericin B (AmB), (SL.5, Table 2) an FDA approved a polyene macrolide antifungal agent is one of them [127,133]. In the field of oncology, liposomal AmB is used in cancer patients with persistent unexplained fever [127,133]. Yeo et al., explained antitumor activity of AmB along with therapeutic efficacy against systemic mycoses [209]. AmB inhibits the transcriptional activity of HIF-1α, not the expression or nuclear translocation. From co-immunoprecipitation assay, it is concluded that after AmB treatment, FIH binding to CAD is increased and interaction between CAD and the C/H1 domain of p300 coactivator is blocked. Therefore, AmB treatment leads to anemia as a result of EPO (Erythropoietin) suppression. EPO is a glycoprotein, which stimulates red blood cell production from bone marrow and it is downstream target of HIF-1α. In the absence of EPO, patients suffer from chronic renal failure. Therefore, there might be nephrotoxicity in prolonged AmB treatment which may exhibit adverse side effects to cancer patients. Therefore, AmB is restricted for use as chemotherapeutic agents in medical oncology.

Polyamides (SL.6, Table 2) are another synthetic sequence specific DNA-binding oligomers, which exert restricted activity of HIF-1α. These are a new type of pyrrole imidazole compounds comprising of two and three aromatic N-methylpyrrole (Py) rings. The architecture of these Polyamides is inspired by the structure of Distamycin and Netropsin. Distamycin and Netropsin are two natural pyrrole-amidine antibiotic compounds originated from actinobacterium *Streptomyces netropsis*. Olenyuk et al., studied the antiangiogenic effects of Polyamides and the competitive inhibition of binding between HIF-1α/ARNT heterodimer to HRE [210]. Polyamides hinder HIF-1α binding to the promoter of proangiogenic protein VEGF and inhibit its transcription. As there are some sequence variations in VEGF splicing variants, Polyamides should be programmed for different sequence binding. Olenyuk and his colleagues designed two different Polyamides specific for 5′-WTWCGW-3′ and 5′-WGGWCW-3′ (where W = A or T) DNA sequences. Therefore, out of these different Polyamides, identification of the best one with prominent anti-cancer activity is dependent on further in vitro and in vivo investigations as well as clinical studies.

Other than these mechanisms, inhibition of nuclear translocation of HIF-1α is another approach for suppressing the transcription of HIF-1α target genes. A good example of an inhibitor that acts using this method is ENMD-1198 (SL.7, Table 2). ENMD-1198 is an analogue of 2-Methoxyestradiol (2ME2) synthesized by Lavallee et al. [211]. 2-Methoxyestradiol (2ME2) is a natural metabolite of estradiol compound with potential anti-HIF-1α activity [211]. Mabjeesh et al., explained that 2ME2 inhibits nuclear translocation of HIF-1α and promotes microtubule disruption [212]. However, according to Moser et al., ENMD-1198 is a tubulin binding agent and it reduces HIF-1α level [213]. Additionally, phosphorylation of MAPK/ERK, PI-3K/AKT and FAK are inhibited by ENMD-1198. Although, 2ME2 and ENMD-1198 have potential to inhibit the transcriptional ability of HIF-1α, further investigations into its anti-HIF-1α activity are required. Although, 2ME2 and ENMD-1198 are in phase II/phase I clinical trials, more investigations to identify their efficiency as HIF-1α inhibitor are needed.

In consideration of different mechanisms of targeting HIF-1α, another approach of HIF-1α regulation through destabilization or degradation by synthetic drugs is discussed in the following section.

#### 4.2.3. HIF-1α Degradation Enhancer

Zeburaline (Zeb) (SL.8, Table 2) is a synthetic DNA methyl transferase inhibitor that also degrades HIF-1α by chemotherapeutic agent, which degrades HIF-1α by enhancing polyubiquitination. Suzuki et al., proved that nuclear localization of HIF-1α is interfered with in oral squamous cell carcinoma (OSCC) by Zeb [214]. It is a synthetic cytidine analogue with antitumor activity. Zeb mediates proteasomal degradation of HIF-1α in a dose-dependent manner whereas the HIF-1α level becomes increased after Zeb treatment in the presence of MG132, a potent proteasome inhibitor [214]. Zebularine affects angiogenesis through various mechanisms. In one case, Zebularine targets the HSP70/HSP90/HIF-1α axis and averts proper folding of HIF-1α leading to premature degradation of HIF-1α. On the other hand, Zebularine prevents nuclear localization of HIF-1α and represses its transcriptional activity. Moreover, Zebularine modulates epigenetic regulation of VEGF and acts as a DNA Methyl Transferase, leading to hypermethylation of proangiogenic genes. Despite having such targets, Zebularine cannot be a novel antiangiogenic drug in clinical use because of its miscellaneous targets and side effects.

Geldanamycin (GA), a natural benzoquinone ansamycin, represses transcription of VEGF, glucose transporters 1 and 3, most of the glycolytic enzymes, and erythropoietin inducing proteasomal degradation of HIF-1α. GA inhibits proper folding of HIF-1α by targeting HSP90. GA blocks ATPase activity of HSP90 by binding its N-terminal ATP binding pocket. Mabjeesh et al., proposed that in GA treatment transcriptional activity of HIF-1α cannot be restored even after use of LCN and MG-132, proteasome inhibitor [215]. In some cases, GA treatment leads to hepatotoxicity in cancer patients [216]. To overcome this toxicity, researchers have synthesized 17-allylamino-17-demethoxygeldanamycin (17-AAG) (SL.9, Table 2), a chemical derivative of Geldanamycin partaking anticancer role in blood, prostate, colonic, hepatic, ovarian, brain, skin, thyroid, renal, and breast cancers, which is supported by several clinical trial as well [127,133]. Geldanamycin and its derivatives are the novel inhibitor with promising pharmacology.

Another HIF-1α inhibitor, Flavopiridol (SL.10, Table 2) is the first semi-synthetic cyclin-dependent kinase (CDK) inhibitor to enter clinical trials [217]. Flavopiridol is a flavonoid compound which possess structural similarities with the alkaloid isolated from Indian endogenous plant *Dysoxylum binectariferum*. Newcomb et al., demonstrated that Flavopiridol mediates proteasomal degradation of HIF-1α [217]. Flavopiridol is used in combination therapy with bortezomib in human leukemia cell lines and induced apoptosis of cell. Proteasome inhibitors potentiate leukemic cell apoptosis induced by the cyclin-dependent kinase inhibitor Flavopiridol through a SAPK/JNK and NF-kβ-dependent process [218]. Along with these, Flavopiridol has a significant effect on regulating cell cytoskeleton, cell cycle components, and cell motility in CSCs. In the case of lung cancer, it was observed that Flavopiridol decreases CD133^high^/CD44^high^ population which is considered to be CSCs [219]. However, it will be fascinating to find the regulation of CD133 and CD44 by HIF-1α in the presence of Flavopiridol as HIF-1α is known to regulate these CD markers [220]. Interestingly, Flavopiridol passed a phase II clinical trial in acute myeloid leukemia, gastric cancer, pancreatic cancer, where it was proved that Flavopiridol is a safe drug with no acute toxicity [127,133,221]. Now, this promising synthetic anti-cancer compound is in phase III clinical trials [222].

Other than the aforesaid mode of action of the HIF-1α targeted synthetic compounds, an alternative mechanism of HIF-1α inhibition is present which targets NF-κB/HIF-1α axis. Figueroa et al., showed that YC-1 represses HIF-1α via inhibition of NF-κB which leads to a significant decrease of EPO expression [224]. Other than this, YC-1 inhibits NF-κB translocation and the mechanistic link between Akt/NF-κB and HIF-1α [225]. Out of the synthetic compounds mentioned in this review, Bortezomib is another one that restricts the degradation of NF-κB inhibitor, i.e., IκBα. Therefore, stabilization of IκBα causes cytoplasmic accumulation and degradation of NF-κB which controls a variety of subsequent oncogenic events, including angiogenesis [206,226,227,228]. Flavopiridol also has similar alternative mode of action such as Bortezomib, which comprises inflammatory response by inhibiting degradation of IκBα. It also inhibits tumor necrosis factor (TNF) mediated NF-κB activation and NF-κB downstream gene expression [229]. Apart from these*,* 17-AAG also has an alternative mode of action, which includes attenuation of the transcriptional activity of NF- κB by targeting HSP90 protein and preventing its coactivator recruitment [230]. Hence, identification of these alternative mode of actions of different synthetic compounds targeting oncogenic pathways such as NF-κB and/PI3K-Akt could be a promising approach for HIF-1α targeted cancer therapy.

Therefore, to follow the particular avenue of HIF-1α targeting, we believe more natural or synthetic compounds are worth exploring to enhance the possibility of successful clinical trial of better anticancer therapeutics. Further studies including combinatorial treatments with natural and synthetic compounds could provide an alternative platform to study HIF-1α targeting therapy.

## 5. Conclusions

Mounting data suggest that HIF-1α is one of the main culprits in the regulation of various hallmarks of cancer. Given the basic role of HIF-1α in the presence of hypoxia, which leads to tumorigenesis, several HIF-1α inhibitors were developed to efficiently treat cancer. Furthermore, the use of natural and synthetic products as HIF-1α inhibitors need more research to reduce their side effects and to decrease their toxicity. Although in clinical studies, until now only few HIF-1α inhibitors have been studied, failure in other cases exerts a major barrier in the development of HIF-1α-targeting anti-cancer therapeutics. As there are very few reports on efficient usage of some HIF-1α targeting drugs against certain type of cancers by clinicians, the pharmacological potential of common and newly proposed HIF-1α inhibitors should be properly documented. In addition, synergistic relationship between the phytochemicals and synthetic HIF-1α inhibitors could prove effective in managing different cancers. Therefore, studies on the synergistic mode of action of natural and synthetic compounds could pave a novel path in HIF-1α targeting anti-cancer therapy.

## Figures and Tables

**Figure 1 molecules-27-05192-f001:**
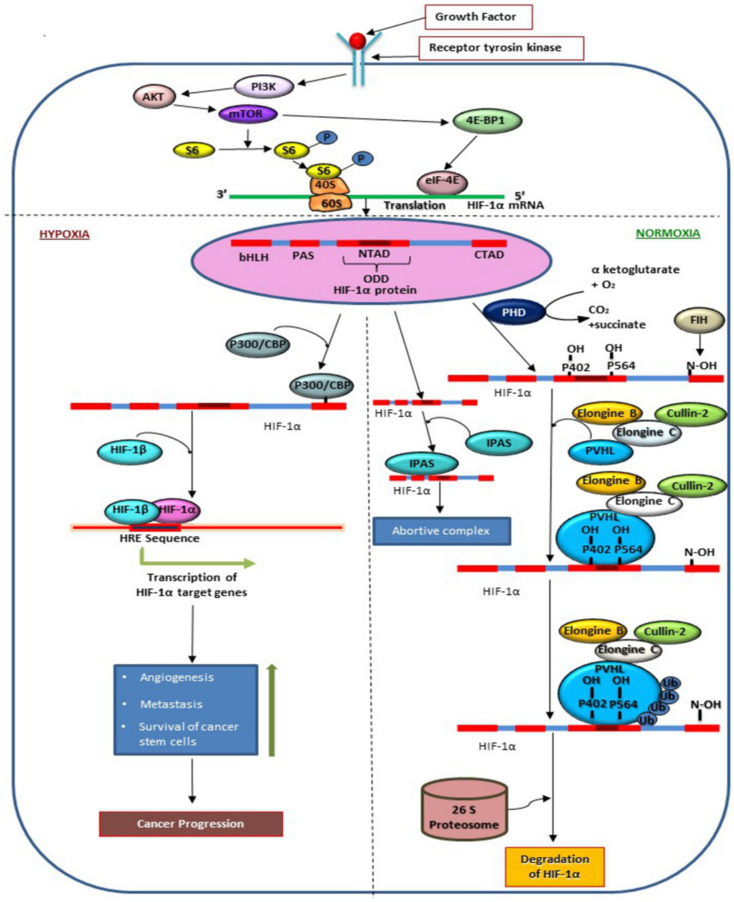
Molecular mechanism of regulation of HIF-1α in hypoxic and normoxic conditions. The figure represents the induction of HIF-1α translation via PI3K/AKT/mTOR pathway. On binding to growth factors at receptor tyrosin kinase PI3K becomes activated which further induces AKT and mTOR pathway activation followed by phosphorylation of S6. HIF-1α synthesis is induced by eIF-4E which binds to HIF-1 α upon activation by 4E-BP1 which is a downstream signaling molecule of mTOR. In the presence of oxygen, Pro-402, Pro-564 in ODD and Asn-803 in CTAD are hydroxylated by PHD and FIH. As represented in the figure, hydroxylation at Asn-803 prevents binding of P300/CBP to HIF-1 α in normoxic conditions, whereas, hydroxylation at Proline residues allow VHL- elongine-C-elongine-B-Cullin-2 complex to bind at ODD of HIF-1α followed ubiquitination of HIF-1α via 26 proteasome. Expression of HIF-1α is also regulated by IPAS, a variant of HIF-3 which binds with HIF-1α to form an abortive complex. In hypoxic conditions, P300/CBP binds at CTAD which prevents degradation of HIF-1α. HIF-1α enters nucleus and forms active transcription factor by binding with HIF-1β in order to transcribe genes for angiogenesis, metastasis, and survival of cancer stem cells in tumor tissue.

**Figure 2 molecules-27-05192-f002:**
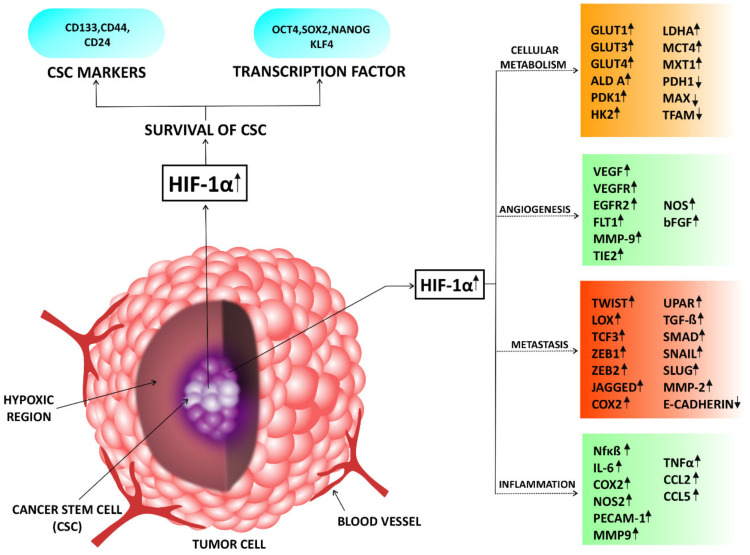
Representation of target genes of HIF-1α involved in tumor progression: Figure represents tumor tissue with hypoxic region at its core. Cancer stem cell (CSC) population resides at the core region. In hypoxic tumor tissue, HIF-1α helps in regulation of genes involved in cellular metabolism [41,42,43,44,45,46,47,48,49,50,51,52,53,54,55,56,57,58,59,60,61,62,63,64,65,66], angiogenesis [67,68,69,70,71,72,73,74,75,76,77,78,79,80,81,82,83,84,85,86], metastasis [87,88,89,90,91,92,93,94,95,96,97,98,99,100,101,102], CSC propagation & maintenance [103,104,105,106,107,108,109,110,111] and cancer inflammation [112,113,114,115,116,117,118,119,120,121,122] which are listed in the figure. Upward arrows (↑) indicate increased expression of the protein and Downward arrows (↓) indicate decreased expression of the protein.

**Figure 3 molecules-27-05192-f003:**
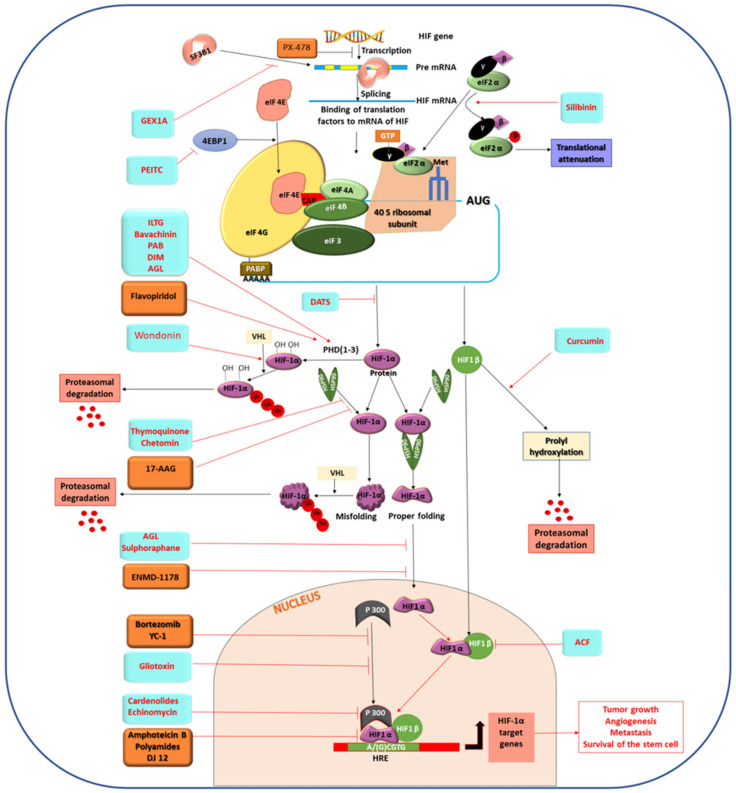
Molecular mechanism of different natural and synthetic compounds targeting HIF-1α and its pathways. The figure represents sky blue and brown boxes that symbolize natural and synthetic compounds, respectively. The illustration denotes GEX1A-mediated inhibition of HIF-1α mRNA splicing by inactivating spliceosome core protein SF3B1. The figure also shows translational attenuates of HIF-1α by silibinin via phosphorylation of α subunit of eIF2. ILTG, Bavachinin, PAB, DIM, AGL, and Flavopiridol-induced prolyl hydroxylation mediated proteasomal degradation of HIF-1α protein is also represented in this figure. Down-regulation of the activity of HIF-1α through proteasomal degradation of its interacting subunit αβ by Curcumin is also demonstrated here. This figure shows the mode of action of Chetomin and Thymoquinone by blocking HSP 90 mediated folding of HIF-1α protein which causes proteasomal degradation of it. Inhibition of nuclear localization of HIF-1α by AGL and Sulforaphane is shown in the figure. Suppression of HIF-1α and HIF-1β heterodimer formation by ACF is also represented. The figure denotes Bortezomib and Gliotoxin dependent inhibition of the interaction between p300 and HIF-1α. Cardenolides, Echinomycin, Amphotericin B, polyamides and DJ12 suppress the binding of HIF-1α/p300 complex to HRE, which are also represented in this figure.

**Table 1 molecules-27-05192-t001:**
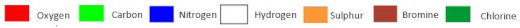
List of various potential natural HIF-1α inhibitors based on their mode of action in different cancers.

Serial No.	Name	Structure	Source	Mode of Action	Effective against Cancer Type	Clinical Trial	Reference
1	Silibinin	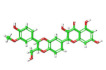	*Silybum marianum*	HIF-1α synthesis blocker	Prostate cancer, cervical cancer, hepatoma, colorectal cancer, nasopharyngeal cancer	Approved	[123,124,125,126,127,128,129,130,131,132,133]
2	Diallyl trisulfide (DATS)	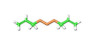	*Allium sativum*	Breast cancer	-----	[127,133,134]
3	Herboxidiene (GEX1A)	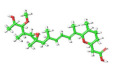	*Streptomyces chromofuscus*	Hepatoma	-----	[127,133,135]
4	Celastrol	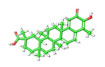	*Tripterygium wilfordii*,*Celastrus regelii*	Glioblastoma	-----	[127,133,136,137,138]
5	PEITC	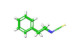	Cruciferous plants	Prostate cancer, human glioma cells, breast cancer	Phase II	[127,133,139,140]
6	Echinomycin	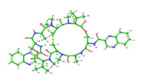	*Streptomyces echinatus*	HIF-1α activity blocker	Breast cancer, acute myeloid leukemia, uterine fibroids	Rejected after phase II trial	[127,133,141,142,143,144,145]
7	Chetomin	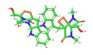	*Chaetomium globosum*	Lung cancer, multiple myeloma	-----	[127,133,146,147,148,149,150]
8	Gliotoxin	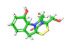	*Gliocladium fimbriatum*	Prostate cancer	-----	[127,133,151,152,153,154]
9	Sulforaphane	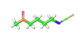	Cruciferous vegetables	Nonmuscle invasive bladder cancer, colon cancer and gastric cancer	Phase II	[127,133,155,156]
10	Acriflavin	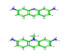	coal tar	Brain cancer, cholangiocarcinoma, ovarian and breast cancer	-----	[127,133,157,158,159,160,161]
11	Emodin	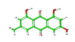	*Rheum palmatum*, *Polygo-nam multiflorum*	Prostate carcinoma	Rejected in clinical trial	[127,133,162,163]
12	Cardenolides	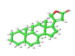	*Calotropis gigantea*	Breast cancer	Entered in clinical trial	[127,133,164,165]
13	DIM (3,3′-Diindolylmethane)	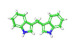	cruciferous vegetables such as broccoli (*Brassica oleracea*), Brussels sprouts, cabbage and kale.	HIF-1α degradation enhancer	Prostate, breast, colon, cervix and pancreas	Phase III	[127,133,166,167,168]
14	Pseudolaric acid B(PAB)	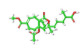	*Pseudolarix kaempferi*	Breast cancer	-----	[127,133,169,170]
15	Bavachinin	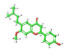	*Psoralea corylifolia*	Human KB carcinoma and HOS osteosarcoma	-----	[127,133,171]
16	Andrographolide	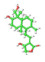	*Andrographis paniculate*	Liver cancer, breast cancer	Phase III	[127,133,172,173]
17	Isoliquiritigenin (ILTG)	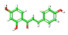	*Glycyrrhiza uralensis, Mongolian glycyrrhiza, Glycyrrhiza glabra.*	Breast cancer	-----	[127,133,174,175]
18	Wondonin	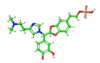	*Poecillastra wondoensis*	Keratinocyte	-----	[127,133,176]
19	Thymoquinone	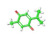	*Nigella sativa*	Renal cancer	Phase II	[127,133,177,178]
20	Curcumin	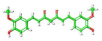	*Curcuma Longa*	Indirect inhibitors of HIF-1α	Breast cancer, pituitary adenoma	Phase II	[127,133,179,180]

**Table 2 molecules-27-05192-t002:**
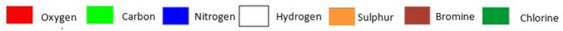
List of various potential synthetic HIF-1α inhibitors based on their mode of action in different cancers.

SerialNo.	Name	Structure	Mode of Action	Effective against Cancer Type	Clinical Trial	Reference
**1**	YC-1	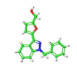	HIF-1α synthesis blocker	Hepatoma, gastric cancer, lung cancer, prostate cancer, pancreatic cancer, Human BladderTransitional Carcinoma	-----	[127,133,195,196,197,198,199,200,201,202]
**2**	PX-478	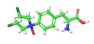	Colon carcinoma, Lung Adenocarcinoma, pancreatic ductal adenocarcinoma	Phase I	[127,133,203,204]
**3**	DJ12	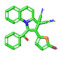	HIF-1α activity blocker	Breast cancer, Melanoma, Renal cancer	-----	[127,133,205]
**4**	Bortezomib	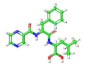	Human hepatoma, multiple myeloma, human embryonic kidney and human multiple myeloma	Approved	[127,133,206,207,208]
**5**	Amphotericin B (AmB)	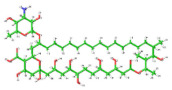		Hepatocellular carcinoma,	Approved	[127,133,209]
**6**	Polyamides	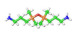	Adenocarcinoma	-----	[127,133,210]
**7**	ENMD-1198 (Analog of 2ME2)	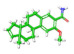	HIF-1α degradation enhancer	Prostate cancer, breast cancer, human hepatocellular carcinoma	-----	[127,133,211,212,213]
**8**	Zebularine	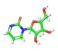	Oral squamous cell carcinoma	-----	[127,133,214]
**9**	17-AAG (Analog of Geldanamycin)	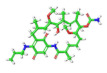	Prostate cancer, renal cell carcinoma, papillary thyroid carcinoma	Phase III	[127,133,215,216]
**10**	Flavopiridol	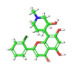	Leukemia, human glioma, neuroblastoma	Phase II	[127,133,217,218,219,220,221,222]

## Data Availability

Not applicable.

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
