# Peer review of "Targeting HIF-1α by Natural and Synthetic Compounds: A Promising Approach for Anti-Cancer Therapeutics Development"

_molecules, 2022, doi:10.3390/molecules27165192_

Round 1

Reviewer 1 Report

Chronic hypoxia triggers a robust reaction of adaptation and repair that is highly conserved along the evolution. In tumors this adaptation plays a relevant role in malignant progression and genesis of metastasis. The modulation/inhibition of HIF1α activity represents a possible strategy to slow/block tumor progression as alternative to the present toxic and inefficient treatments. Therefore, this review can be useful to many readers and is welcome.

However, I have a substantial concern and a few minor observations before the acceptance of this manuscript:

a) Hypoxia and prolonged activation of HIF1α are strictly related with a proinflammatory reparative response and activation of NFkB, not only in the proinflammatory cells, but also in tumor cells itself, generating the malignant phenotype. Both HIF1a and NFkB are sensitive to the redox status and to the damage production. A number of genes related to invasion and metastasis (such as MMPs, new adhesive molecules, receptors for chemokines, and others) are controlled by both transcription factors. Stemness genes, TERT, growth factors and proliferation-related genes and energy metabolism key enzymes may be also controlled, at least to some extent, by both HIF1α and NFkB. This imply that inhibitors and modulator of NFkB must be also taken in account for this anti-progression strategy.

My question: why in this review NFkB is totally ignored

Minor:

1-Abstract+Introduction:

  • Provide a reference about the 70 mm distance between blood vessels and cells for the hypoxic oxygen restriction. 200-300 mm is the distance generally experienced/cited in different tissues, including tumors.
  • The major targets of ROS are lipids (lipoperoxidation) !

2-Structure and regulation of HIF1a:

  • This paragraph can be drastically shortened citing a large number of good reviews already published.

3-Role of HIF1a in cancer progression:

  • The activation and contribution to the proinflammatory-reparative response is missing: Toll-like receptors, inducible enzymes (iNOS, COX-2, HOX1, etc.), MMps, acute-phase proteins, chemokine receptors, adhesive molecules and countereceptors for homing are missing. The last three are relevant to understand the polarized migration and homing of cancer cells and the preferential distribution of metastasis.

4-Natural and synthetic compounds:

  • To be noted: many natural compound are effective on both HIF1a and NFkB
  • On-market off-label drugs (such as digitalis derivatives and acriflavin) should be discussed and cited (especially effective in ongoing clinical trials for many tumors).
  • Clinical trials of combination of HIF1a+NFkB inhibitors should be also discussed.

Author Response

Comment: Chronic hypoxia triggers a robust reaction of adaptation and repair that is highly conserved along the evolution. In tumors this adaptation plays a relevant role in malignant progression and genesis of metastasis. The modulation/inhibition of HIF1α activity represents a possible strategy to slow/block tumor progression as alternative to the present toxic and inefficient treatments. Therefore, this review can be useful to many readers and is welcome.

However, I have a substantial concern and a few minor observations before the acceptance of this manuscript:

  1. a) Hypoxia and prolonged activation of HIF1α are strictly related with a proinflammatory reparative response and activation of NFkB, not only in the proinflammatory cells, but also in tumor cells itself, generating the malignant phenotype. Both HIF1a and NFkB are sensitive to the redox status and to the damage production. A number of genes related to invasion and metastasis (such as MMPs, new adhesive molecules, receptors for chemokines, and others) are controlled by both transcription factors. Stemness genes, TERT, growth factors and proliferation-related genes and energy metabolism key enzymes may be also controlled, at least to some extent, by both HIF1α and NFkB. This imply that inhibitors and modulator of NFkB must be also taken in account for this anti-progression strategy.

My question: why in this review NFkB is totally ignored?

Response: We wish to thank the reviewer for his very constructive comments and suggestions, these helped to strengthen the content of the review. As per reviewers’ suggestion we have incorporated the portions mentioning the relation between HIF-1α and NFkB in context of oncogenic progression (Page No. 9, Line no. 26-36 & Page No. 12. Line no. 2-28). We have also mentioned the mode of action of some HIF-1α targeting natural and synthetic compounds involving NFkB as potent modulator (Page no. 24, Line no. 1- 10). We hope the given emphasis on this part will satisfy the reviewer and help the readers in better understanding of the review.

Minor:

1-Abstract+Introduction:

Provide a reference about the 70 mm distance between blood vessels and cells for the hypoxic oxygen restriction. 200-300 mm is the distance generally experienced/cited in different tissues, including tumors.

Response: As directed by the reviewer we have provided reference stating the fact that the 70 µm distance between blood vessels and cells for the hypoxic oxygen restriction. The reference number “3” is newly introduced to the manuscript in support of this information. The details of the reference is “Vaupel, P.; Thews, O.; Hoeckel, M. Treatment Resistance of Solid Tumors: Role of Hypoxia and Anemia. Med. Oncol. Northwood Lond. Engl. 2001, 18, 243–259, doi:10.1385/MO:18:4:243”. All other references are changed accordingly in the manuscript.

The major targets of ROS are lipids (lipoperoxidation) !

Response: The reviewer correctly stated that lipids are one of the major targets of ROS. We are thankful to the reviewer for indicating the same. We have incorporated the word “lipids” to the sentence “Generation of ROS triggers damage of cellular biomolecules like lipids, proteins, DNA and RNA” in “Introduction” section of the manuscript (Page No. – 2, Line No. - 16 ).

2-Structure and regulation of HIF1a:

This paragraph can be drastically shortened citing a large number of good reviews already published.

Response: As per reviewers’ suggestion we have shortened the paragraph (Page no. 4-5) and rearranged the references accordingly. All the changes are tabulated in the “Specific modifications in the manuscript” of the rebuttal letter.

3-Role of HIF1a in cancer progression:

The activation and contribution to the proinflammatory-reparative response is missing: Toll-like receptors, inducible enzymes (iNOS, COX-2, HOX1, etc.), MMps, acute-phase proteins, chemokine receptors, adhesive molecules and countereceptors for homing are missing. The last three are relevant to understand the polarized migration and homing of cancer cells and the preferential distribution of metastasis.

Response: We wish to thank the reviewer for the valuable comments. As mentioned by the reviewer we have incorporated the portions related to the activation and contribution to the proinflammatory-reparative response in the newly introduced part “Role of HIF-1α in cancer-related inflammatory response” in page no. 12 from line no. 30 to 33.

4-Natural and synthetic compounds:

To be noted: many natural compound are effective on both HIF1a and NFkB

On-market off-label drugs (such as digitalis derivatives and acriflavin) should be discussed and cited (especially effective in ongoing clinical trials for many tumors).

Clinical trials of combination of HIF1a+NFkB inhibitors should be also discussed.

Response: We like to thank the reviewer for providing these insights. We have rewritten and incorporated the alternative mode of action of the compounds which are dealing with both HIF-1α and NFkB as the targets (Page no. 24, Line no. 1-10 & Page no. 29, Line no. 19-34 ). As suggested by the reviewer we have also discussed the efficacy of these compounds for other diseases, as well as, the clinical trial status of these compounds for different cancers. These parts are added to the paragraphs related the description of each compounds.

Reviewer 2 Report

This is a reasonably comprehensive review of HIF-1α, as well as the status of therapeutic development targeting the protein. The authors have nicely summarized the current understanding of the role that HIF-1α plays in regulating cellular processes, including primarily its role in cancer development and progression. Furthermore, the manuscript offers insight into the therapeutic moieties in development as anticancer agents. 

This reviewer finds no faults in the scientific teachings of the manuscript; however, the document needs thorough editing to correct its numerous grammatical and typographical errors. In its current form, the manuscript is readable, but its flow and progression are severely impeded by incorrect sentence structure and missing articles.

Additionally, this reviewer has the following suggestions:

  • First, the authors should normalize all chemical structures to the journal's standards, i.e., single size with appropriate (atomic) color.
  • Structures should be numbered to allow for easy reference back to the table when discussed in the text.
  • Column 2 in Tables 1 and 2 should either be removed or include more relevant information – ex. listing diallyl trisulfide as an organosulfur compound serves no purpose, nor does listing 17-AAG as a synthetic compound in a table of synthetic HIF-1α inhibitors. 
  • Et al. should not be written as et.al
  • Please check the spelling of target (taeget) and Cullin (Callin) in Figure 1.

Overall, this reviewer believes the authors have written a valuable review that will be widely useful to the field and recommends publication after the above issues are addressed.

Reviewer 3 Report

The present review article present insights into the complexity of cellular metabolism, metastasis, tumor angiogenesis and survival of cancer stem cells induced by hypoxia-inducible factor-1α (HIF-1α). The authors discuss about natural and synthetic compounds as HIF-1α inhibitors, which have the potential to accelerate anticancer drug discovery. Also introduces the mode of action of these compounds for a better understanding of the chemical leads that could be useful as cancer therapeutics in the future. The following technical flaws/comments/concerns should be addressed before publication consideration in “Molecules”.

Major Drawbacks

  1. Lack of Novelty, originality, and presentation of obsolete topic: There are review articles published previously on the same topic and few examples are as follows. Also, describe how this article different from others.

Natural compounds and the hypoxia-inducible factor (HIF) signaling pathway. September 2009Biochimie 91(11-12):1347-58.

Natural products as potent inhibitors of hypoxia-inducible factor-1α in cancer therapy. Chinese Journal of Natural Medicines. Volume 18, Issue 9, September 2020, Pages 696-703.

Anti-Cancer Activity of Phytochemicals Targeting Hypoxia-Inducible Factor-1 Alpha. Int. J. Mol. Sci. 2021, 22(18), 9819

Natural Product-Based Inhibitors of Hypoxia-Inducible Factor-1 (HIF-1). Curr Drug Targets. 2006 Mar; 7(3): 355–369.

Targeting hypoxia-inducible factor 1 (HIF-1) signaling with natural products toward cancer chemotherapy. The Journal of Antibiotics volume 74, pages687–695 (2021).

HIF-1: structure, biology and natural modulators. Chinese Journal of Natural Medicines. Volume 19, Issue 7, July 2021, Pages 521-527.

HIF-1α is a Potential Molecular Target for Herbal Medicine to Treat Diseases. 16 November 2020 Volume 2020:14 Pages 4915—4949.

  1. This draft is written more likely a book chapter than a review article since there are many statements or sentences where inevitable references are missing. Few examples are as follows,
  2. Introduction, page 2; However, other than involvement in angiogenic progression HIF-1α…surrounding tumor tissue and invasion (line 40-44). Role of HIF-1α in metastasis…… Here references with respect to metastasis and EMT are missing.
  3. Introduction, page 2; Various studies reported that hypoxia…... to adopt a different way to survive (line 48-50). Role of HIF-1α promoting enrichment of cancer stem cells (CSCs). Please cite the missing references.
  4. Introduction, page 2; line 48. “Various studies reported that hypoxia leads …………tumor cells”. References are missing for this claim.
  5. Introduction, page 3; line 12. “HIF-1α is critical for…….attenuated tumorigenesis”.  References are not cited in the text.
  6. Introduction, page 3; line 16-19. “Several drugs are already…………to block HIF-1α pathways. References are inevitable.
  7. Introduction, page 6: Role of HIF-1α in Cancer Progression.  In response to both hypoxic stress… propagation and maintenance (line,15-21). Reference are missing.
  8. Introduction, page 6 and 7 (3.1. Role of HIF-1α in Cellular Metabolism). The whole paragraph should be read carefully and rewrite with only main ideas related and cite proper references accordingly.
  9. Page 8; Figure 2. Representation of target genes of HIF-1α involved in tumor progression. Cite the missing reference in the figure legend.
  10. Page 9 (line, 2-4); Loss of HIF-1α also disrupts…. of solid tumor in vivo. Line 5-6, Another study showed that …...signaling pathway in neuroblastoma cell line.  Cite missing references in the text.
  11. Page 9, line 18. “Besides, several studies reported ……… HIF-1α”. References are inevitable.
  12. Page 9, line, 21: “Quintero et.al reported that……….in squamous cell carcinoma”. Reference is missing.
  13. Page 9, line, 24: “HIF-1α also regulates angiogenesis via PI3/AKT signaling pathway………………….. HIF-1α”. Reference is missing.
  14. Page 9, line, 30: “It is reported that………………….. HIF-1α”. Reference is missing.
  15. Page 9, line, 46: “various studies revealed that………………….. inducing EMT”. References are inevitable.
  16. Page 10. Lines 39. Cancer stem cells (CSCs) are …... with the production of CSC markers. Reference is missing.
  17. Page 10. Line 51. In case of glioma, breast cancer ……... after treatment. References are not cited in the text.

  1. All the authors should read the manuscript at least two times to check the language or vague descriptions and the technical flaws such as grammatical, typos as well as repetition.
  2. The content is too long with unnecessary information and the flow of ideas is disconnected. 

Author Response

The present review article present insights into the complexity of cellular metabolism, metastasis, tumor angiogenesis and survival of cancer stem cells induced by hypoxia-inducible factor-1α (HIF-1α). The authors discuss about natural and synthetic compounds as HIF-1α inhibitors, which have the potential to accelerate anticancer drug discovery. Also introduces the mode of action of these compounds for a better understanding of the chemical leads that could be useful as cancer therapeutics in the future. The following technical flaws/comments/concerns should be addressed before publication consideration in “Molecules”.

Major Drawbacks

  1. Lack of Novelty, originality, and presentation of obsolete topic: There are review articles published previously on the same topic and few examples are as follows. Also, describe how this article different from others.

      Natural compounds and the hypoxia-inducible factor (HIF) signaling pathway. September

      2009Biochimie 91(11-12):1347-58.

      Natural products as potent inhibitors of hypoxia-inducible factor-1α in cancer therapy. Chinese

      Journal of Natural Medicines. Volume 18, Issue 9, September 2020, Pages 696-703.

      Anti-Cancer Activity of Phytochemicals Targeting Hypoxia-Inducible Factor-1 Alpha. Int. J. Mol.     

      Sci. 2021, 22(18), 9819

     Natural Product-Based Inhibitors of Hypoxia-Inducible Factor-1 (HIF-1). Curr Drug Targets. 2006

     Mar; 7(3): 355–369.

     Targeting hypoxia-inducible factor 1 (HIF-1) signaling with natural products toward cancer   

     chemotherapy. The Journal of Antibiotics volume 74, pages687–695 (2021).

     HIF-1: structure, biology and natural modulators. Chinese Journal of Natural Medicines. Volume

     19, Issue 7, July 2021, Pages 521-527.

     HIF-1α is a Potential Molecular Target for Herbal Medicine to Treat Diseases. 16 November 2020   

     Volume 2020:14 Pages 4915—4949.

Response: We wish to thank the reviewer for his comments and suggestions. Even though the reviewer is concern about the novelty, originality and presentation of the topic, we would like to state that, to the best of our knowledge there is no such review article which is written on the current scenario of research related to HIF-1α targeting compounds, specifically on both the natural and synthetic compounds. All the articles mentioned by the reviewer is only based on the natural products, most of which are discussing the effect of different compounds on different subtypes of HIF as well as HIF-1. On the other hand, in our review we have precisely discussed about HIF-1α and both the natural and synthetic compounds targeting it. According to us, inclusion of paragraphs like “Role of HIF-1α in Cancer stem cell proliferation and maintenance” or discussion of HIF-1α targeting natural and synthetic compounds by categorizing them as HIF-1α synthesis blocker, HIF-1α activity blocker, HIF-1α degradation enhancer or Degrader of HIF-1α interacting HIF subunits make it more novel and relevant. We believe that this review has the capability to attract broad readership among the researchers working on cancer, hypoxia, HIF1α and/or drug discovery.

  1. This draft is written more likely a book chapter than a review article since there are many statements or sentences where inevitable references are missing. Few examples are as follows,

Response: We like to thank the reviewer again for his valued suggestions. We have incorporated several references in the manuscript as suggested by the reviewer. The details of the newly introduced references are mentioned in “References” section of the manuscript as well as in the “Specific modifications in the manuscript” section of the rebuttal letter. We hope this will suffice to satisfy the reviewer.

  1. Introduction, page 2; However, other than involvement in angiogenic progression HIF-1α…surrounding tumor tissue and invasion (line 40-44). Role of HIF-1α in metastasis…… Here references with respect to metastasis and EMT are missing.

Response: As suggested by the reviewer we have introduced new references (Ref. No. 15-17) related to metastasis and EMT after the mentioned line.

  1. Introduction, page 2; Various studies reported that hypoxia…... to adopt a different way to survive (line 48-50). Role of HIF-1α promoting enrichment of cancer stem cells (CSCs). Please cite the missing references.

Response: As per reviewers’ suggestion we have incorporated new references (Ref. No. 18-19) for the mentioned portions.

  1. Introduction, page 2; line 48. “Various studies reported that hypoxia leads …………tumor cells”. References are missing for this claim.

Response: As suggested by the reviewer we have introduced new reference (Ref. No. 18-19) to the mentioned line.

  1. Introduction, page 3; line 12. “HIF-1α is critical for…….attenuated tumorigenesis”.  References are not cited in the text.

Response: As per reviewers’ recommendation we have mentioned proper references (Ref. No. 19-26) for the sentence.

  1. Introduction, page 3; line 16-19. “Several drugs are already…………to block HIF-1α pathways. References are inevitable.

Response: As suggested by the reviewer we have introduced new reference (Ref. No. 27) to the mentioned line.

  1. Introduction, page 6: Role of HIF-1α in Cancer Progression.  In response to both hypoxic stress… propagation and maintenance (line,15-21). References are missing.

Response: As per reviewers’ recommendation we have incorporated suitable references ((Ref. No. 39,40) for the mentioned lines.

  1. Introduction, page 6 and 7 (3.1. Role of HIF-1α in Cellular Metabolism). The whole paragraph should be read carefully and rewrite with only main ideas related and cite proper references accordingly.

Response: We have rewritten the paragraph entitled “Role of HIF-1α in Cellular Metabolism” and also modified the “Introduction” portion as suggested by the reviewer. We hope it will suffice the need of the readers.

  1. Page 8; Figure 2. Representation of target genes of HIF-1α involved in tumor progression. Cite the missing reference in the figure legend.

Response: All the missing references are freshly introduced to the figure legends of Figure 2.

  1. Page 9 (line, 2-4); Loss of HIF-1α also disrupts…. of solid tumor in vivo.Line 5-6, Another study showed that …...signaling pathway in neuroblastoma cell line.  Cite missing references in the text.

Response: As per reviewers’ recommendation we have incorporated suitable reference (Ref. No. 78) for the mentioned lines.

  1. Page 9, line 18. “Besides, several studies reported ……… HIF-1α”. References are inevitable.

Response: As per reviewers’ recommendation we have incorporated suitable references (Ref. No. 81,82) for the mentioned lines.

  1. Page 9, line, 21: “Quintero et.al reported that……….in squamous cell carcinoma”. Reference is missing.

Response: As suggested by the reviewer we have introduced new reference (Ref. No. 81) to the mentioned line.

  1. Page 9, line, 24: “HIF-1α also regulates angiogenesis via PI3/AKT signaling pathway………………….. HIF-1α”. Reference is missing.

Response: As mentioned by the reviewer we have introduced new references (Ref. No. 83,84) to the sentence.

  1. Page 9, line, 30: “It is reported that………………….. HIF-1α”. Reference is missing.

Response: As per reviewers’ recommendation we have mentioned proper reference (Ref. No. 85) for the sentence.

  1. Page 9, line, 46: “various studies revealed that………………….. inducing EMT”. References are inevitable.

Response: As suggested by the reviewer we have introduced new reference (Ref. No. 88,89) to the mentioned line.

  1. Page 10. Lines 39. Cancer stem cells (CSCs) are …... with the production of CSC markers. Reference is missing.

Response: As per reviewers’ recommendation we have mentioned proper reference (Ref. No. 19) for the sentence.

  1. Page 10. Line 51. In case of glioma, breast cancer ……... after treatment. References are not cited in the text.

Response: As suggested by the reviewer we have introduced new reference (Ref. No. 103) to the mentioned line.

  1. All the authors should read the manuscript at least two times to check the language or vague descriptions and the technical flaws such as grammatical, typos as well as repetition.

Response: We are thankful to the reviewer as his comments helps us to improve the manuscript and make it flawless. We have corrected all grammatical and typographical errors as well as repetition of the manuscript. We appreciate his concern in this regard.

  1. The content is too long with unnecessary information and the flow of ideas is disconnected. 

Response: We have rechecked all the contents of this article and introduced necessary changes to the manuscript to ensure the proper flow of ideas. All the changes in the manuscript are tabulated in the “Specific modifications in the manuscript” section of the rebuttal letter.

Round 2

Reviewer 1 Report

Changes required improved the manuscript.

It can be accepted in the present form.

Reviewer 3 Report

The authors have satisfactorily addressed most of my comments and concerns raised in the previous review, and I recommend the manuscript for publication.